# LANGUAGE AGENTS MEET CAUSALITY – BRIDGING LLMS AND CAUSAL WORLD MODELS

John Gkountouras[1], Matthias Lindemann[2], Phillip Lippe[3], Efstratios Gavves[3,4], and Ivan Titov[2,1]

[1]Institute for Logic, Language and Computation (ILLC), University of Amsterdam
[2]Institute for Language, Cognition and Computation (ILCC), University of Edinburgh
[3]QUVA Lab, University of Amsterdam
[4]Archimedes/Athena RC, Greece
`i.gkountouras@uva.nl, m.m.lindemann@sms.ed.ac.uk,`
`p.lippe@uva.nl, egavves@uva.nl, ititov@inf.ed.ac.uk`

## ABSTRACT

Large Language Models (LLMs) have recently shown great promise in planning and reasoning applications. These tasks demand robust systems, which arguably require a causal understanding of the environment. While LLMs can acquire and reflect common sense causal knowledge from their pretraining data, this information is often incomplete, incorrect, or inapplicable to a specific environment. In contrast, causal representation learning (CRL) focuses on identifying the underlying causal structure within a given environment. We propose a framework that integrates CRLs with LLMs to enable causally-aware reasoning and planning. This framework learns a causal world model, with causal variables linked to natural language expressions. This mapping provides LLMs with a flexible interface to process and generate descriptions of actions and states in text form. Effectively, the causal world model acts as a simulator that the LLM can query and interact with. We evaluate the framework on causal inference and planning tasks across temporal scales and environmental complexities. Our experiments demonstrate the effectiveness of the approach, with the causally-aware method outperforming LLM-based reasoners, especially for longer planning horizons.

## 1 INTRODUCTION

Large Language Models (LLMs) have emerged as powerful tools for a wide range of tasks, from natural language understanding to complex problem-solving (Brown et al., 2020; Radford et al., 2019; Liu et al., 2023b). Recent work has explored the use of LLMs as action agents for planning and reasoning tasks, showing promising results in improving task-specific, downstream performance (Ahn et al., 2022; Hao et al., 2023; Huang et al., 2023). These approaches primarily rely on the model's ability to extract common-sense causal information stated in its training data (Zečević et al., 2023). While LLMs can reflect general beliefs and correlations, this information may be incomplete, incorrect, or inapplicable in specific environments. This poses challenges for LLMs in novel or complex situations, particularly in dynamic environments where accurate modeling of action consequences is crucial (Valmeekam et al., 2023; Kambhampati et al., 2024).

Causal representation learning (CRL) aims to identify the underlying causal structure of data (Schölkopf et al., 2021). By separating and identifying latent causal factors, CRL enables models to reason about the effects of interventions and counterfactuals. Recent theoretical work provides justification for causal representation learning, showing it is necessary for achieving strong robustness guarantees in AI systems (Richens & Everitt, 2024). While CRL can model complex causal mechanisms, applying it to real-world environments with visual complexity remains challenging. Recent advancements in CRL (Lippe et al., 2022; 2023) have begun to tackle this problem in simulated environments. These developments open up new possibilities for enhancing AI systems,

---

Project page: https://j0hngou.github.io/LLMCWM/.

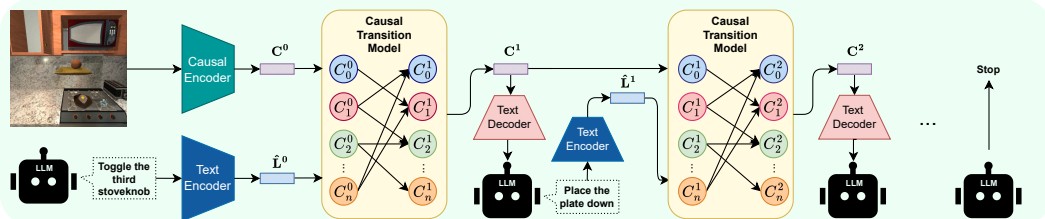

Figure 1: Overview of a single rollout in the proposed planning pipeline. The causal encoder, implemented using a CRL model, maps the high-dimensional state representation (image) to its fundamental constituents—the causal variables. During planning, the LLM agent samples a proposed action, which is then encoded by the text encoder. The causal transition model uses both the disentangled latent representation of the image and the encoded action to simulate the next state based on its learned causal mechanisms. The process iterates until the planning algorithm terminates, with the causal model autoregressively operating in the latent space.

including LLMs. Although CRL does not directly address all LLM limitations, it can significantly enhance their capabilities in specific domains. Our work builds upon these advancements, integrating CRL with language models to improve their performance on causal inference and planning tasks.

We introduce a framework that combines CRL with language models to enable causally-aware reasoning and planning in interactive environments. CRL provides LLMs with a structured, causal understanding of the environment, reasoning about interventions and their consequences during planning. The causal world model – akin to a simulator but learned rather than predefined – allows the LLM to evaluate multiple possible futures before taking action and thereby guides its decisions. Conversely, LLMs offer a flexible interface for interacting with the causal world model, allowing for more intuitive planning and reasoning that can leverage the LLM's commonsense knowledge.

Furthermore, this work investigates using text to represent actions in the context of CRL-based world modeling. Text-based action representations provide a flexible and intuitive way to describe actions, making them more suitable for generalist agents operating in diverse environments. Moreover, annotating frame sequences with natural language descriptions is often easier than exhaustively enumerating every possible action in an environment, which can be intractable for complex domains.

We consider a setting with interleaved sequential observations in image format and corresponding action descriptions at each timestep. This setup takes inspiration from real-world scenarios where an agent might receive visual input (e.g., from a camera) along with descriptions of actions taken (e.g., from system logs or human annotations). For example, in a robotic manipulation task, the dataset might consist of a series of images showing the robot's workspace, paired with descriptions like *"The gripper shifted slightly to the right."* or *"The object was grasped and placed on the worktop."* We assume no prior knowledge of the causal factors or the causal mechanisms between them. The agent can only observe the effects of its actions from the images and does not require explicit information about which specific variables or factors in the causal model it is affecting. Our method builds on BISCUIT (Lippe et al., 2023), a CRL framework, to create a flexible causally-aware world model from the sequence of observations and action descriptions, which is then used for planning in environments.

The key contributions of our work are as follows:

- The first framework integrating CRL with LLMs to enable causally-aware reasoning and planning in interactive environments.
- An exploration of text-based action representations for CRL and demonstration of their effectiveness in data-scarce regimes, showing improved data efficiency in learning causal representations.
- Demonstration of the framework's effectiveness on a set of reasoning and planning tasks across both static and dynamic environments.

Our experiments focus on simple environments, using existing CRL methods that are sufficiently advanced for our use case. While these environments are still relatively simple, they represent the current frontier of causal representation learning. As more powerful CRL methods become available, they can be integrated into our framework, scaling it to more complex, realistic scenarios.

## 2 RELATED WORK

**Causal Representation Learning**   Causal representation learning aims to identify the underlying causal variables and their relations from high-dimensional observations (Schölkopf et al., 2021). In the most general setting, the latent causal variables may not be uniquely identifiable (Locatello et al., 2019a; Hyvärinen & Pajunen, 1999). Many approaches rely on assumptions or additional knowledge about the causal structure, such as constraining the observation function (Buchholz et al., 2023; Squires et al., 2023; Ahuja et al., 2023; Zhang et al., 2023; Kivva et al., 2022; Lachapelle et al., 2023), sparse graphical structures (Khemakhem et al., 2020; Liu et al., 2022; 2024; Lachapelle & Lacoste-Julien, 2022; Lachapelle et al., 2024), having multiple views (Xu et al., 2024; Yao et al., 2024a; von Kügelgen et al., 2021; Brehmer et al., 2022; Locatello et al., 2020), or supplementary supervision labels (Yang et al., 2020; Komanduri et al., 2022; Locatello et al., 2019b). Recent advancements have explored CRL for temporal environments, in which agent-level actions like in reinforcement learning are used to learn the causal structure of the environment (Lippe et al., 2022; 2023; Nalmpantis et al., 2023). In particular, our work leverages BISCUIT (Lippe et al., 2023), a CRL framework that learns causal representations with realistic agent-focused assumptions, requiring only a small set of labeled causal variables for the final mapping *after* causal representation learning, without their interactions or causal graphs.

**World Models and Causal Integration**   World models predict the consequences of actions and have been extensively used in reinforcement learning (Ha & Schmidhuber, 2018). Recent work has focused on object-centric world models (Greff et al., 2017; Steenkiste et al., 2018; Watters et al., 2019) and the integration of graph neural networks for modeling transitions (Battaglia et al., 2016; 2018; Kipf et al., 2018). However, attempts to integrate causality into world models have been limited. Some approaches, such as CoPhyNet (Baradel et al., 2020), consider counterfactual scenarios but rely on direct supervision of object positions or place constraints on unobserved variables (Li et al., 2020). Our work aims to learn a causal world model relying only on images and textual annotations [1] but capable of reasoning about actions across state transitions, while also being able to be interacted with by a language model.

**Large Language Models, Causality, Planning and Reasoning**   There has been much work exploring the use of LLMs as action agents for planning and reasoning tasks, showing promising results (Ahn et al., 2022; Hao et al., 2023; Huang et al., 2023). Various methodologies have been developed to make use of LLMs for agent planning. These include task decomposition for breaking complex tasks into subtasks (Wei et al., 2022; Yao et al., 2023; Shen et al., 2024), multi-plan selection for generating and choosing optimal plans (Yao et al., 2024b; Wang et al., 2022), external module-aided planning (Liu et al., 2023a; Guan et al., 2023), reflection and refinement via self-evaluation and improvement (Shinn et al., 2024; Gou et al., 2024; Madaan et al., 2024), and memory-augmented planning for decision making (Zhang et al., 2024; Zhong et al., 2024). While LLMs have shown impressive performance in reasoning, tool usage, planning, and instruction-following, challenges remain in addressing hallucinations, plan feasibility, and tractability in complex, multi-step planning scenarios (Valmeekam et al., 2023; Kambhampati et al., 2024; Kambhampati, 2024). Theoretical work on robustness under distribution shifts in unmediated decision tasks (where the decision does not influence the utility) establishes a connection between causal understanding and robustness (Richens & Everitt, 2024). A better approximation of the underlying causal model generally translates to more robust agents, implying that world models should be causality-aware (Gupta et al., 2024).

## 3 BACKGROUND AND SETUP

To enable LLMs to perform causally-aware reasoning and planning in interactive environments, we leverage CRL methods to build a causal world model (CWM). The CWM provides LLMs with a structured understanding of the environment, allowing them to reason about interventions and their consequences during planning.

In this section, we provide an overview of CRL in temporal causal graphs, which is foundational to our framework. We discuss how CRL can learn latent causal representations from sequences of observations and actions, setting the stage for integrating these representations with LLMs.

---

[1]Except for a few labels needed to map from the latent representation to human-interpretable language.

### 3.1 Causal Representation Learning in Temporal Causal Graphs

CRL aims to uncover the latent causal variables and the underlying causal structure. In temporal settings, we consider sequences of high-dimensional observations $\{\mathbf{X}^t\}_{t=0}^T$, where $\mathbf{X}^t \in \mathbb{R}^D$, and actions $\{\mathbf{R}^t\}_{t=1}^T$, where $\mathbf{R}^t \in \mathbb{R}^E$. Actions $\mathbf{R}^t$ can represent, for example, the coordinates of the locations where the interactions occurred (Lippe et al., 2023). The true causal variables $\{\mathbf{C}^t\}_{t=0}^T$, where $\mathbf{C}^t \in \mathbb{R}^K$, are **unobserved**. Furthermore, a deterministic observation model is assumed, often represented as $\mathbf{X}^t = g(\mathbf{C}^t)$, where $g : \mathbb{R}^K \to \mathbb{R}^D$ is an injective function mapping causal variables to observations.

Instead of directly modeling causal variables, CRL relies on latent state representations. It estimates a function $f : \mathbb{R}^D \to \mathbb{R}^M$ [2] that maps observations $\mathbf{X}^t$ to latent representations $\mathbf{z}^t = f(\mathbf{X}^t)$. The goal is to ensure that each dimension $z_i^t$ of $\mathbf{z}^t$ corresponds to a causal variable $C_i^t$ in $\mathbf{C}^t$ up to a transformation decided by the identifiability class of the causal model. Specifically, it aims to achieve this disentanglement using only $\{\mathbf{X}^t\}_{t=0}^T$ and $\{\mathbf{R}^t\}_{t=1}^T$.

### 3.2 Generative Model

The temporal CRL framework is often modeled as a generative process that describes how observations are produced from underlying latent state representations and actions. At each time step $t$, the state $\mathbf{z}^t$ evolve according to a transition model influenced by actions $\mathbf{R}^t$, and generate observations $\mathbf{X}^t$. Assuming a first-order Markov process, the conditional likelihood of the observed data $\{\mathbf{X}^t\}_{t=0}^T$ given actions $\{\mathbf{R}^t\}_{t=1}^T$ is expressed as

$$p\big(\{\mathbf{X}^t\} \mid \{\mathbf{R}^t\}\big) = \int p(\mathbf{z}^0) \prod_{t=1}^T p_\omega(\mathbf{z}^t \mid \mathbf{z}^{t-1}, \mathbf{R}^t)\, p_g(\mathbf{X}^t \mid \mathbf{z}^t)\, \mathrm{d}\mathbf{z}, \tag{1}$$

where $p(\mathbf{z}^0)$ is the prior distribution over the state. The *transition model* term $p_\omega(\mathbf{z}^t \mid \mathbf{z}^{t-1}, \mathbf{R}^t)$ models the state dynamics, capturing how the states evolve over time and how intervening actions influence them. The observation model $p_g(\mathbf{X}^t \mid \mathbf{z}^t)$ describes how the states generate the observations, which in our case will be done with the deterministic function $g$.

The marginalization in Eq. (1) renders the objective intractable. A standard approach to address this is to optimize the corresponding Evidence Lower Bound (ELBO) by assuming a Gaussian distribution for the transition dynamics and the standard Gaussian for the prior, using the reparameterization trick to enable efficient optimization (Kingma & Welling, 2013).

### 3.3 Identifiability Guarantees in BISCUIT

There is nothing in the objective of Eq. (1) itself that guarantees the model will identify the causal variables from the observations. In BISCUIT (Lippe et al., 2023), the CRL framework we adopt, identifiability arises from two key assumptions: (1) each causal variable has a distinct 'interaction pattern,' meaning that the effect of $\mathbf{R}^t$ on $\mathbf{z}^t$ is mediated by a latent binary mask, and (2) these interaction patterns vary over time. The first assumption is enforced by using a structured model family to model the transition $p_\omega(\mathbf{z}^t \mid \mathbf{z}^{t-1}, \mathbf{R}^t)$. We incorporate this component from BISCUIT in our approach. These assumptions together ensure that causal variables are uniquely identifiable from the observed data. For a more detailed discussion on the assumptions, theoretical guarantees, and the structure of the transition model, we refer the reader to the original paper (Lippe et al., 2023).

## 4 Building a Causal World Model from Causal Representations

To integrate the CRL model with LLMs, we construct a Causal World Model (CWM) that takes actions in text format and states in image format and produces state representations in natural language. The CWM builds on BISCUIT to model the environment's dynamics, with the CWM's encoder and decoder components (see Figure 2) responsible for translating states and actions to and from natural language. BISCUIT ensures identifiability and causal structure recovery, which enables reliable predictions of the effects of actions/interventions, as demonstrated by our experiments in Section 6.

---

[2] Since we do not normally have a priori information of the number of causal variables, we set $M \gg K$.

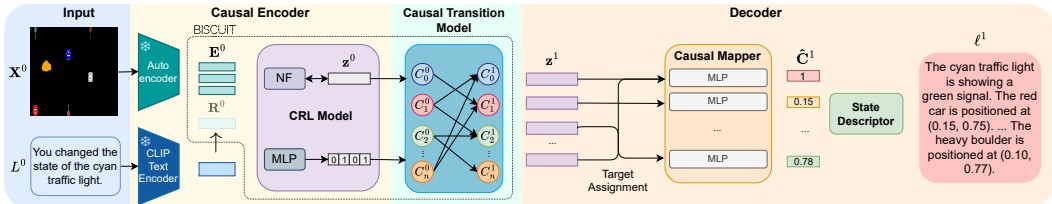

Figure 2: Illustration of the first roll-out step with the Causal World Model. The image $\mathbf{X}^0$ and action description $L^0$ are encoded into initial latent representations. The CRL module then disentangles these representations and the transition model predicts the next state. The causal mapper transforms the disentangled causal representation of the next state into the estimated causal variables $\hat{\mathbf{C}}^1$. Finally, the state descriptor $s$ generates a natural language description $\ell^1$ of the next state. For subsequent steps, the model can autoregress in the latent space using the previously predicted $\mathbf{z}$, bypassing the autoencoder and normalizing flow, enabling efficient multi-step inference and planning.

Although we utilize BISCUIT, our approach should be compatible with any CRL frameworks that can provide disentangled causal representations (i.e., (Nalmpantis et al., 2023; Lachapelle et al., 2024; Yao et al., 2022)).

## 4.1 LANGUAGE GROUNDING MODULES

To integrate the CRL model with LLMs, we introduce architectural components that transform the CRL model into a world model with a language interface. This section outlines the new components we introduce, enabling the model to process image states and text inputs, and produce text outputs.

**Language-Based Action Representations** We replace the action encoding $\mathbf{R}^t$ in the CRL framework with a language-based representation $\mathcal{L}_e(L^t)$, where $\mathcal{L}_e$ embeds a natural language description $L^t$. This is implemented using an encoder-only language model (Reimers & Gurevych, 2019) with a trainable head, replacing the original action encodings in the CRL framework's *transition model* $\mathbf{R}^t = \mathcal{L}_e(L^t)$ (see also Section 4.2).

**Decoder** The decoder $\mathcal{G}$ comprises two parts: the causal mapper and the state description generator. The **causal mapper** $m_\theta$ extracts causal variables $\mathbf{C}$ from the learned disentangled representations $\mathbf{z}$. It first identifies which latent dimensions $z_i$ are most predictive for each causal variable $C_j$, then learns to perform the actual mapping. The **state description generator** $s$ maps the estimated causal variables $\hat{\mathbf{C}}$ to $\ell$, a natural language description of the state. Detailed implementations of these components are provided in Appendix G and H respectively.

## 4.2 PARAMETER ESTIMATION AND INFERENCE

In this section, we explain the estimation process for all model components and detail how the resulting model is applied during inference. We use the GridWorld environment as a running example to illustrate the process, though the same methodology applies to any environment.

**Estimation: Causal Encoder and Transition Model** To estimate the model, we use image pairs $\{\mathbf{X}^t\}$ and corresponding action descriptions $\{L^t\}$ in natural language, for example, "you toggled the cyan traffic light" or "you moved the blue car". We first train an autoencoder to compress high-dimensional observations $\mathbf{X}_t$ into lower-dimensional latent representations $\mathbf{E}_t = e_\psi(\mathbf{X}_t)$, in which, however, the causal variables will still be entangled. Then, analogously to Eq. 1, the conditional likelihood of the encoded observations $\{\mathbf{E}^t\}_{t=0}^T$ given action descriptions $\{L^t\}_{t=1}^T$ is given by

$$p(\{\mathbf{E}^t\} \mid \{L^t\}) = \int p(\mathbf{z}^0) \prod_{t=1}^{T} p_\omega(\mathbf{z}^t \mid \mathbf{z}^{t-1}, L_e(L^t)) \, p_\phi(\mathbf{E}^t \mid \mathbf{z}^t) \, d\mathbf{z}, \qquad (2)$$

where $p_\omega$ is the transition model, structured as in BISCUIT in order to satisfy the identifiability guarantees, $p_\phi(\mathbf{E}^t \mid \mathbf{z}^t)$, is the observation model, and $p(\mathbf{z}^0)$ is the prior distribution over the initial latents, assumed to be the standard Gaussian. The invertible mapping $f_\phi : \mathbb{R}^M \to \mathbb{R}^M$ is a

normalizing flow (NF) that transforms the autoencoder's states $\{\mathbf{E}^t\}$ into a new, structured latent space $\{\mathbf{z}^t\}$, while identifying and separating the causal variables. As the NF is invertible, $f_\phi$ and its Jacobian also yield the term $p_\phi(\mathbf{E}^t \mid \mathbf{z}^t)$ in the generative model. Similar to BISCUIT, the ELBO is formulated and optimized using the reparameterization trick (Kingma & Welling, 2013).

The full causal encoder that maps an observation $\mathbf{X}^t$ to the causal state $\mathbf{z}^t$ is expressed as $\mathcal{E} := f_\phi \circ e_\psi$, where $e_\psi$ is the encoder part of the autoencoder, and $f_\phi$ is the NF. While we assume perfect reconstruction capability for the autoencoder, a common assumption in CRL (Kivva et al., 2022; Lachapelle et al., 2023; Brehmer et al., 2022; Lachapelle et al., 2024, inter alia), this component can be replaced with stronger visual encoders as they become available, without affecting the framework's core functionality.

This framework builds upon the BISCUIT architecture, maintaining the same structure for the autoencoder, normalizing flow (RealNVP (Dinh et al., 2016)), and transition model. However, we introduce an important modification in the action representation. While BISCUIT used coordinate-based action encodings (as described in Section 5.1), our work incorporates language-based action representations through $\mathcal{L}_e(L^t)$ in the transition model to enable our model to process natural language action descriptions.

**Estimation: Decoder**    We train the causal mapper $m_\theta$ using a small set of annotated but not necessarily ordered images where the true causal variables $\mathbf{C}$ and their values are known. The training pairs consist of $(\mathbf{z}, \mathbf{C})$, where $\mathbf{z}$ is the output of our causal encoding pipeline and $\mathbf{C}$ are the corresponding ground truth causal variables. In GridWorld, these variables include positions of vehicles and obstacles, and states of traffic lights. For instance, the causal mapper might learn that dimensions $z_1$, $z_3$, and $z_7$ are most predictive for the "blue car x-position" ($C_0$), and then train a specific predictor for $C_0$ using only these relevant dimensions.

The state description generator $s$, typically a rule-based system, maps the estimated causal variables to human-interpretable outputs. For example, it might transform position and state variables into a description like "The blue car is at (2,3), the cyan traffic light is green". The full decoder is expressed as $\mathcal{G} := s \circ m_\theta$.

**Inference Process**    During inference, the model sequentially processes new GridWorld images through these components:

$$\mathbf{z}^t = \mathcal{E}(\mathbf{X}^t) = (f_\phi \circ e_\psi)(\mathbf{X}^t),$$
$$\mathbf{z}^{t+1} \sim p_\omega\big(\mathbf{z}^{t+1} \mid \mathbf{z}^t, \mathcal{L}_e(L^t)\big),$$
$$\ell^{t+1} = \mathcal{G}(\mathbf{z}^{t+1}) = (s \circ m_\theta)(\mathbf{z}^{t+1}).$$

This process transforms raw input into interpretable state descriptions of the next state, facilitating interaction with language models for reasoning and planning tasks. Notably, the transition model operates solely in the disentangled latent space, without dependency on the high-dimensional observations $\mathbf{X}^t$. This enables efficient multi-step inference through autoregression, allowing for long-term planning and reasoning without the need to decode back to the observation space at each step.

This entire process relies solely on the sequence of observations and action descriptions, without requiring explicit information about which specific variables or factors are being affected. By introducing the language-based action encoder and decoding into natural language, we create a framework that is inherently suited for language-based causal reasoning in complex environments. The algorithm to perform inference is provided in Appendix M.

## 5    EXPERIMENTAL SETUP

We evaluate our framework using two distinct environments: a dynamic $8 \times 8$ GridWorld and a static 3D-rendered kitchen (AI2-THOR) (Kolve et al., 2017). The GridWorld is dynamic, meaning the environment state can change even without agent actions, while the iTHOR kitchen is static, changing only in response to agent interventions. Our experiments focus on three key aspects: the effectiveness of text-based action representations, causal inference, and planning. Both environments feature various objects with causal variables representing their states and positions. Detailed descriptions of the environments are provided in Appendices A and B.

For each environment, we generated multiple datasets for training, evaluation, and in-context learning. The data generation process involves initializing the environment state and performing random, valid actions. Specific details about dataset sizes, in-context learning example generation, and self-evaluation reward generation for planning tasks are described in Appendix D.

## 5.1 ACTION REPRESENTATIONS

We investigate three action representation modalities:

1. **Coordinate-based (CB):** Encoding of 2D pixel coordinates indicating the position where the interaction was performed. For example, a click at position $(2, 3)$ is transformed into a higher-dimensional representation using high-frequency sinusoidal functions.
2. **Text-based (TB):** Natural language descriptions expanded using a PCFG (e.g., "you toggled the bright cyan traffic light"), then encoded through an encoder-only text embedder.
3. **Hybrid (HB):** Concatenation of coordinate-based and text-based representations.

We hypothesize that the text-based action encoding is a) semantically richer, providing more information for the same or less effort to annotate the data, b) more flexible, enabling a language-based interface suitable for a generalist agent, and c) more robust, meaning that paraphrases or equivalent descriptions of the same action can still work with our model even if it was not specifically trained on them. This last point is crucial, as the LLM used at inference may deviate in its action description style from what was seen during training.

## 5.2 BASELINE

Our baseline uses the world model component from the Reasoning via Planning (RAP) methodology (Hao et al., 2023). This language model-based world model predicts the next state given the current state $s_t$, chosen action $a_t$, and context $c$:

$$s_{t+1} \sim p_{\text{LM}}(s_{t+1} \mid s_t, a_t, c).$$

The baseline constructs a prompt at runtime that includes the environment description and dynamics, current state representation, chosen action, two relevant in-context learning (ICL) examples, and instructions for predicting the next state. This approach leverages the language model's pretrained knowledge while adapting to the specific task and environment dynamics. We ensure the relevance of the ICL examples by providing examples that match the current action and the object it is applied to.

We use LLaMA 3 (8B) (Dubey et al., 2024) as the planning agent quantized to 6 bits in the exl2 format. We chose RAP+LLaMA3 as the baseline for its simplicity and effectiveness, providing a fair point of comparison to assess the benefits of integrating causal representation learning. This allows us to isolate the impact of causal understanding in an otherwise comparable framework, though our approach could integrate with alternative search algorithms such as LLM-MCTS (Zhao et al., 2024).

## 6 EXPERIMENTS AND DISCUSSION

### 6.1 EVALUATION OF TEXT-BASED ACTION REPRESENTATIONS

In this experiment, we demonstrate the effectiveness of representing actions in natural language for learning causal representations. We assess the induced state variables $\mathbf{z}$ by comparing them to ground-truth causal variables. Note that the model's decoder is not evaluated in these experiments.

We train our causal world model using each action modality (**CB**, **TB**, **HB**) across different subsample percentages of the training dataset, focusing on the low-data regime. Given sufficient data, models yield practically identical results across all 3 modalities but obtaining data in non-simulated environments is typically expensive. Performance is assessed using a standard CRL metric: $R^2$ scores for the permutation $\pi$ that maximizes the diagonal of the $R^2$ matrix between learned latent variables and true causal variables. This approach accounts for the fact that we learn causal variables up to permutation. Each experiment uses 3 seeds with distinct subsamples. A more comprehensive explanation of the training of the components of the CRL models used is presented in Appendix E.

Table 1: $R^2$ scores for action representations. **CB**: Coordinate-based, **TB**: Text-based, **HB**: Hybrid. $100\%$ stands for $10^6$ image states.

| Action Type | Subsample Percentage | | | | | | 100% |
|---|---|---|---|---|---|---|---|
| | **0.3%** | **0.5%** | **0.7%** | **1.0%** | **1.2%** | **1.5%** | |
| **CB** | **0.392** ±0.000 | 0.366±0.000 | 0.424±0.001 | 0.457±0.004 | 0.472±0.004 | 0.548±0.022 | 0.987 |
| **TB** | 0.374±0.000 | 0.362±0.000 | 0.399±0.000 | **0.470**±0.012 | **0.495**±**0.014** | **0.603**±**0.003** | 0.990 |
| **HB** | **0.392**±**0.000** | **0.433**±**0.001** | **0.460**±**0.000** | 0.461±0.007 | 0.490±0.010 | 0.539±0.011 | **0.991** |

Table 2: $N$-step causal inference accuracies for the causal world model and the RAP (Hao et al., 2023) world model across different environments and step lengths.

| Steps | iTHOR | | | GridWorld | | | | |
|---|---|---|---|---|---|---|---|---|
| | **1** | **2** | **4** | **1** | **2** | **4** | **6** | **8** |
| Causal Model | **0.824** | **0.680** | **0.630** | **0.954** | **0.922** | **0.829** | **0.797** | **0.758** |
| RAP World Model | 0.482 | 0.285 | 0.110 | 0.391 | 0.220 | 0.085 | 0.045 | 0.005 |

Table 1 presents the results of our action representation experiments for the GridWorld environment in the low-data regime. Our results demonstrate that incorporating text into action representations (**TB** and **HB**) is Pareto-optimal in GridWorld; **TB** and **HB** perform at least as well as the coordinate-based representation, especially in low-data regimes. In extremely low-data scenarios ($0.3\%$ - $0.7\%$), the hybrid approach consistently outperforms both **CB** and **TB**. As data increases ($1.0\%$ - $1.5\%$), **TB** shows competitive performance while providing natural alignment with LLM interfaces.

These findings support our hypothesis: action encodings including text are as effective as or superior in uncovering causal variables to coordinate-based representations, particularly when data is scarce. We use the **TB** model trained using the entire dataset ($10^6$ examples) in the subsequent experiments.

## 6.2 CAUSAL INFERENCE PERFORMANCE

Our causal inference experiments evaluate both world models' ability to perform $1$-step and $N$-step causal inference, i.e., predict the effects of actions (interventions) on the environment. In the $1$-step case, given the current state and an action, the model predicts the new state. For $N$-step causal inference, we provide a sequence of actions and only the starting state and the world model applies each action to its previous prediction in a sequence. This differs from planning in that it focuses on the effect of a *given* sequence of actions rather than *finding* actions to reach a goal. The evaluation methodology is presented in Appendix K.

Table 2 presents accuracies of causal inference for both models across different environments and step lengths. The causal world model consistently outperforms the baseline across all scenarios. In GridWorld, it maintains high accuracy ($75.8\%$) even for $8$-step inference, while the baseline's performance drops nearly to $0$. The performance in iTHOR, while lower than in GridWorld, still shows a substantial improvement over the baseline.

The higher overall performance on GridWorld can be attributed to its simpler action space, object space, and causal graph, despite its dynamic nature. The baseline's lower performance in GridWorld compared to iTHOR may be due to the lack of visual input, which is less natural for language models in an artificial environment.

Table 3 provides a detailed breakdown of the causal inference performance for specific actions and objects, based on the extended 1-step dataset of 3000 samples. In iTHOR, the causal world model excels at ToggleObject and OpenObject actions ($95.7\%$ and $92.6\%$ accuracy), while struggling more with PutObject and PickupObject actions ($50.6\%$ and $43.1\%$ accuracy). This discrepancy likely stems from the following; first, we model the three-dimensional coordinates as independent random variables while, in reality, they are dependent. Second, we model interventions using binary variables to estimate whether we performed an intervention or not. Performance could be improved by injecting inductive bias towards the continuous, three-dimensional nature of the underlying variable. However, this requires task specialization within the model and we chose to keep the proposed framework task-

Table 3: Causal inference accuracies for action categories in iTHOR and GridWorld environments. CWM: Causal World Model, RAP (Hao et al., 2023): RAP World Model Baseline.

| iTHOR Environment | | | GridWorld Environment | | |
|---|---|---|---|---|---|
| **Action Category** | **CWM** | **RAP** | **Action Category** | **CWM** | **RAP** |
| ToggleObject | **0.957** | 0.466 | Change Light State | **0.986** | 0.300 |
| OpenObject | **0.926** | 0.339 | No Action | **0.985** | 0.456 |
| NoOp | **0.962** | 0.710 | Move | **0.928** | 0.408 |
| PutObject | **0.506** | 0.100 | | | |
| PickupObject | 0.431 | **0.692** | | | |

agnostic. The baseline model shows a different pattern, performing better on NoOp and PickupObject actions but struggling with PutObject actions.

In GridWorld, the causal world model demonstrates high accuracy across all action types, with particularly strong performance in changes to the state of the lights and no-action scenarios. The baseline model shows lower performance across the board, with its best performance on the No Action category.

These results highlight the causal world model's superior ability to reason about causal relationships, maintaining strong performance across different temporal scales, environments, and action types.

## 6.3 PLANNING

**Methodology** The planning experiments assess the model's ability to generate a sequence of actions to transform an initial state into a goal state. This involves exploring multiple possible action sequences and evaluating their effectiveness in reaching the goal. Unlike causal inference, planning requires considering long-term consequences and optimizing for a specific objective.

Our framework adapts the Reasoning via Planning (RAP) methodology, with a key distinction: we employ a separate causal world model alongside a language model agent, rather than using a single language model for both roles. We use the same LLM as for the baseline planning agent (LLaMA 3).

The planning works as follows: The LLaMA 3 agent proposes possible actions based on the current state. The world model then simulates the actions' outcomes, predicting subsequent states. The agent then evaluates each state-action pair's quality and picks an action resulting in a new state. This cycle repeats, exploring multiple reasoning paths before converging on a final solution. For all $N$-step experiments, we use a search tree depth of $N + 2$. We use a modified version of the RAP-MCTS algorithm, presented in Appendix N.

**Actions** In **Gridworld**, there are three actions to toggle traffic lights (one per light) and one to perform no action. In **iTHOR**, we dynamically generate 10-15 possible actions, depending on the initial state. During planning, the models use their internal representations to determine possible actions. During evaluation, we use the external simulator (the same one used to generate the data) to execute the plan proposed by the agent. If an invalid action is proposed, during evaluation we default to performing no action.

**Reward Design** In line with the RAP methodology, we rely on the LLM's ability to judge the current state in relation to the goal. The **Intuition reward** is the unnormalized log probability of actions generated by the language model, given the current state and few-shot demonstrations. The **Self-evaluation reward** is the log probability of the token "good" when asking the model to evaluate whether the proposed action is correct, given the current state and few-shot demonstrations.

We avoid using **percentage-of-goals-reached** rewards to maintain generality and applicability to problems that are not easily divisible into subgoals or subtasks. This choice ensures that our method remains applicable to a wide range of problems, including those where intermediate progress toward the goal is difficult to quantify and/or may not correlate directly with overall success.

### 6.3.1 PLANNING RESULTS AND DISCUSSION

Table 4 presents the planning results for both models across different environments and step lengths.

Table 4: Planning results for the causal world model and RAP (Hao et al., 2023) across different environments and step lengths. The best performing method for each metric is highlighted in green.

| Environment | Steps | Causal World Model | | | RAP Baseline | | |
|---|---|---|---|---|---|---|---|
| | | Success Rate ↑ | Avg. Steps (Success) ↓ | Avg. Steps (Failure) ↓ | Success Rate ↑ | Avg. Steps (Success) ↓ | Avg. Steps (Failure) ↓ |
| iTHOR | 2 | **0.58** | **1.78** | **3.36** | 0.25 | 4.00 | 4.00 |
| | 4 | **0.44** | **2.14** | **5.43** | 0.11 | 6.00 | 6.00 |
| GridWorld | 2 | **0.95** | **1.92** | 3.20 | 0.20 | 2.00 | **3.19** |
| | 4 | **0.73** | **2.71** | **5.27** | 0.11 | 3.27 | 5.48 |
| | 6 | **0.46** | **3.65** | **7.72** | 0.08 | 5.50 | 7.72 |
| | 8 | **0.42** | **4.62** | **9.76** | 0.06 | 7.00 | 9.93 |

The causal world model consistently outperforms the baseline in both environments:

- **Success Rates:** The causal model achieves significantly higher success rates, particularly in longer planning horizons. In iTHOR, it more than doubles the baseline's success rate for 2-step planning ($0.58$ vs $0.25$) and quadruples it for 4-step planning ($0.44$ vs $0.11$).
- **Efficiency:** For successful trajectories, the causal model takes fewer steps on average to reach the goal state, indicating more efficient planning.
- **Scalability:** While both models show decreased performance as the number of steps in the ground truth increase, the causal model degrades more gracefully. In GridWorld, it maintains a $0.42$ success rate for 8-step planning, compared to the baseline's $0.06$.
- **Consistency:** Both models perform better in GridWorld compared to iTHOR, likely due to the lower complexity and more constrained action space. However, the causal model shows more consistent performance across both environments.

An interesting observation is the sub-$N$ performance in $N$-step planning scenarios. This phenomenon arises from two key factors in our experimental design which renders the parameter $N$ an upper bound of the steps needed to achieve the goal state. In the static iTHOR environment, some actions can negate others (e.g., toggling the toaster twice is equivalent to performing no action), allowing for shorter paths to the goal state. In addition to this phenomenon, in the dynamic GridWorld environment, the inherent movement of entities (e.g., cars moving when facing a green light) can sometimes lead to the goal state in fewer steps than the upper bound. This sub-$N$ performance highlights our models' ability to find efficient paths to the goal state, often outperforming the original trajectories used to generate the planning problems.

The performance improvements observed in our experiments can be attributed to the CRL world model's higher accuracy in predicting future states. Both our method and the baseline use identical state representations, with consistent text formatting maintained across the in-context learning examples. This uniformity ensures that the language model's reasoning and planning processes are influenced primarily by the accuracy of the underlying world model. Therefore, the superior planning performance of our method highlights the effectiveness of integrating CRL for more accurate state predictions, which directly benefits the downstream reasoning tasks.

# 7 CONCLUSION

In this work, we introduced a framework that integrates causal representation learning with language models, enabling causally-aware reasoning and planning in interactive environments. Our approach combines the structured causal understanding of CRL with the flexible interface of language models, demonstrating superior performance in causal inference and planning tasks across two environments. The causal world model consistently outperforms baselines, showing improved accuracy, efficiency, and scalability as task complexity increases. Our exploration of text-based action representations reveals potential advantages in low-data regimes, suggesting implications for more flexible and generalizable AI systems. While our current experiments focus on relatively simple environments, the framework is designed to extend to more complex scenarios as CRL and search methods advance. Future work could explore applications to real-world environments, improve the interpretability of learned causal world models and develop techniques independent of labeled causal variables.

## REPRODUCIBILITY STATEMENT

For reproducibility, we publish the code and models to integrate Causal Representation Learning (CRL) with Language Models (LLMs), as well as the scripts to generate data sets used in our experiments, on our code repository: https://github.com/j0hngou/LLMCWM/. All models were implemented using PyTorch (Paszke et al., 2019) and PyTorch Lightning (Falcon & The PyTorch Lightning team, 2019). Detailed hyperparameters and dataset descriptions are provided in Section 5 and Section 6, with further details in Appendices C, F, D, E, K, M, and N.

In terms of computational resources, all experiments were performed on NVIDIA A100 GPUs. Training the autoencoders takes approximately 1 to 2 days. Jointly training the normalizing flows and the language heads takes around 0.5 to 1 hour.

## ACKNOWLEDGMENTS

The authors thank the Netherlands Organization for Scientific Research (NWO) for their support (VICI grant VI.C.212.053).

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

## A  GRIDWORLD ENVIRONMENT

The gridworld environment is a dynamic environment of size $H \times H$, where $H \in \mathbb{N}$ denotes both the height and width of the grid. The top left corner of the grid is defined to be $(0, 0)$. The environment consists of $C$ underlying causal variables that interact based on the actions taken by the agent and the dynamics of the environment. The environment contains three types of entities: vehicles $v \in V$, obstacles $o \in O$, and traffic lights $tl \in TL$. Each entity has a fixed corresponding attribute, implemented as a color, which differentiates it from other objects within the same entity class.

The traffic lights are positioned in the grid, and each vehicle is facing a specific traffic light. The positions of the traffic lights are fixed and immutable, with coordinates $(x_{tl}, y_{tl})$, where $x_{tl}, y_{tl} \in \{0, 1, \ldots, H - 1\}$. Each traffic light has a state $s_{tl} \in \{\text{red, green}\}$. The obstacles have positions $(x_o, y_o)$ in the grid, where $x_o, y_o \in \{0, 1, \ldots, H - 1\}$, and these positions can only change through interventions performed on them. The vehicles have positions $(x_v, y_v)$ in the grid, where $x_v, y_v \in \{0, 1, \ldots, H - 1\}$, and an orientation $\theta_v \in \{\text{up, down, left, right}\}$. The vehicle positions change according to the following dynamics:

Let $v$ be a vehicle at position $(x_v, y_v)$ with orientation $\theta_v$, associated with a traffic light $tl$ at position $(x_{tl}, y_{tl})$. We say that the vehicle $v$ is facing the traffic light $tl$ if and only if one of the following conditions is satisfied:

1. $\theta_v = \text{up}$ and $x_v = x_{tl}$ and $y_v > y_{tl}$
2. $\theta_v = \text{down}$ and $x_v = x_{tl}$ and $y_v < y_{tl}$
3. $\theta_v = \text{left}$ and $y_v = y_{tl}$ and $x_v > x_{tl}$
4. $\theta_v = \text{right}$ and $y_v = y_{tl}$ and $x_v < x_{tl}$

If the vehicle $v$ is facing the traffic light $tl$, it will move forward to the cell $(x'_v, y'_v)$ at the next timestep if and only if all of the following conditions are satisfied:

1. The traffic light $tl$ has a state of green, i.e., $s_{tl} = \text{green}$.
2. There are no obstacles in the cell $(x'_v, y'_v)$, i.e., $\nexists\, o \in O : (x_o, y_o) = (x'_v, y'_v)$.
3. There are no traffic lights in the cell $(x'_v, y'_v)$, i.e., $\nexists\, tl \in TL : (x_{tl}, y_{tl}) = (x'_v, y'_v)$.
4. The cell $(x'_v, y'_v)$ is within the grid boundaries, i.e., $0 \leq x'_v < H$ and $0 \leq y'_v < H$.

The new position $(x'_v, y'_v)$ is determined by the vehicle's current position $(x_v, y_v)$ and orientation $\theta_v$ as follows:

$$
(x'_v, y'_v) = \begin{cases} (x_v, y_v - 1) & \text{if } \theta_v = \text{up} \\ (x_v, y_v + 1) & \text{if } \theta_v = \text{down} \\ (x_v - 1, y_v) & \text{if } \theta_v = \text{left} \\ (x_v + 1, y_v) & \text{if } \theta_v = \text{right} \end{cases} \tag{3}
$$

**Interventions**  The intervention process follows a specific sequence: first, a step in the environment dynamics is executed; then, an intervention is applied; finally, a snapshot of the resulting state is captured. Interventions can modify traffic light states, alter obstacle positions, or move a vehicle forward. Spatial interventions on obstacles and vehicles are constrained to single-cell displacements; for obstacles, the direction is stochastic, while for vehicles, it is deterministically forward. Vehicle intervention is further constrained by the absence of obstacles or traffic lights in the target cell, adherence to environment boundaries, and the corresponding traffic light displaying a red signal. A no-operation (NOOP) intervention is also permissible. This tripartite sequence—environmental progression, intervention, and state documentation—constitutes a complete intervention cycle. These interventions correspond to regime variables $\mathbf{R}^t$, which are then represented using natural language.

**Causal Variables**  The causal variables in the gridworld environment are the positions of the vehicles $(x_v, y_v)$, the positions of the obstacles $(x_o, y_o)$, and the states of the traffic lights $s_{tl}$.

## B IThor Kitchen Environment - Embodied AI

The iTHOR (Kolve et al., 2017) kitchen environment is based on the FloorPlan10 dataset, featuring a static 3D-rendered kitchen. The robot's position remains fixed in front of the kitchen counter. The environment consists of $C$ underlying causal variables that interact based on the actions taken by the agent. The environment contains three types of entities: movable objects $m \in M$, fixed interactive objects $f \in F$, and receptacles $r \in R$. Movable objects include a plate with a potato and an egg. Fixed interactive objects comprise a microwave, stoves, cabinet, and toaster. Receptacles include the counter, microwave (when open), and pan (for the egg). Each object has a state $s_o \in S_o$, where $S_o$ is the set of possible states for object $o$. For binary state objects (e.g., microwave, cabinet), $S_o =$ open, closed or active, inactive. For movable objects, $S_o$ includes their position $(x_m, y_m, z_m)$ in the 3D space and a binary pickup state. The set of possible actions $A$ includes:

- ToggleObject($o$): For $o \in \{$microwave, stoves, toaster$\}$
- OpenObject($o$): For $o \in \{$microwave, cabinet$\}$
- PickupObject($m$): For $m \in M$
- PutObject($m, r$): For $m \in M, r \in R$
- MoveObject($m$): For $m \in M$
- NoOp: No action performed

The availability of actions depends on the current state of objects. For example:

$$\text{ToggleObject(microwave) is valid iff } s_{\text{microwave}} = \text{closed} \tag{4}$$

$$\text{OpenObject(microwave) is valid iff } s_{\text{microwave}} = \text{inactive} \tag{5}$$

The regime variable $\mathbf{R}^t \in [0, 1]^2$ represents the normalized click-location on the image to select the object for interaction. Let $I_o$ be the set of pixels belonging to object $o$ in the current frame. Then:

$$\mathbf{R}^t = \frac{1}{H \times W} \cdot (x, y), \text{ where } (x, y) \sim \text{Uniform}(I_o) \tag{6}$$

where $H$ and $W$ are the height and width of the frame respectively. The causal variables $\mathbf{C} = C_1, ..., C_C$ in this environment correspond to the states and positions of objects. Binary state variables (e.g., Cabinet-Open, Microwave-Active) take values in $0, 1$, while position variables (e.g., Egg-Pos-x) take continuous values in $[0, 1]$, normalized to the environment's dimensions. Observations are generated as high-resolution images $X^t \in \mathbb{R}^{512 \times 512 \times 3}$, then downsampled to $X'^t \in \mathbb{R}^{256 \times 256 \times 3}$ using bilinear interpolation.

## C Text-Based Action Representation Generation

### C.1 GridWorld Environment

For the GridWorld environment, we implement a probabilistic context-free grammar (PCFG). The PCFG includes:

- A set of adjectives $A_o$ for each object type $o \in O = \{$traffic light, vehicle, obstacle$\}$
- A set of action modifiers $M$
- A set of action verbs $V_a$ for each action type $a \in A = \{$move, turn, change state$\}$

Let $\mathcal{C} : \mathbb{R}^3 \rightarrow \Sigma_c$ be a function mapping RGB values to a finite set of color names $\Sigma_c$. For each object $o$ with RGB value $r_o$, we compute its color name as $c_o = \mathcal{C}(r_o)$. The generation process for an action $a$ on object $o$ can be formalized as:

$$D(a, o) = m \cdot v_a \cdot \text{the} \cdot adj_o \cdot c_o \cdot o \tag{7}$$

where $m \sim P(M)$, $v_a \in V_a$, $adj_o \sim P(A_o)$, and $P(\cdot)$ denotes the probability distribution defined by the PCFG. *Example:* Consider an action to move a blue car to the right. Let $r_o = (0, 0, 255)$, $\mathcal{C}(r_o) =$ "blue", $m =$ "skillfully", $v_a =$ "moved", and $adj_o =$ "sleek". The generated description would be:

$$D(\text{move right, car}) = \text{"You skillfully moved the sleek, blue car to the right."} \tag{8}$$

## C.2 ITHOR ENVIRONMENT

For the iTHOR environment, we define a mapping function $f : A \times O \to \Sigma$, where $A$ is the set of possible actions, $O$ is the set of objects, and $\Sigma$ is the set of all possible strings over the alphabet. Let $T_a : A \to V$ be a function that maps actions to verb phrases, and $T_o : O \to \Sigma^*$ be a function that maps objects to descriptive phrases. The generation process for an action $a$ on object $o$ can be expressed as:

$$f(a, o) = \text{You} \cdot T_a(a) \cdot T_o(o) \tag{9}$$

*Example:* For the action of opening a microwave, let $a = \text{OpenObject}$ and $o = \text{Microwave}$. Assume $T_a(\text{OpenObject}) = $ "adjusted" and $T_o(\text{Microwave}) = $ "the microwave's door". The generated description would be:

$$f(\text{OpenObject}, \text{Microwave}) = \text{"You adjusted the microwave's door."} \tag{10}$$

## C.3 TOKENIZATION AND INTEGRATION

Let $\tau : \Sigma^* \to \mathbb{N}^k$ be a tokenization function that maps a string to a sequence of $k$ token indices. For a generated description $d$, we compute its tokenized representation as:

$$t = \tau(d) \tag{11}$$

The tokenized representations are then padded or truncated to a fixed length $l$, resulting in the final representation $t' \in \mathbb{N}^l$. For a trajectory of actions $a_1, ..., a_n$ on objects $o_1, ..., o_n$, we generate a sequence of tokenized descriptions $t'_1, ..., t'_n$, where:

$$t'_i = \text{pad}(\tau(D(a_i, o_i)), l) \text{ for GridWorld} \tag{12}$$

$$t'_i = \text{pad}(\tau(f(a_i, o_i)), l) \text{ for iTHOR} \tag{13}$$

# D   DATA GENERATION AND PREPARATION

For each environment, we generated multiple datasets as shown in Table 5.

Table 5: Dataset specifications for each environment

| Dataset | Size | Description |
|---------|------|-------------|
| Training | 10000 trajectories of 100 steps | Used for model training |
| Validation | 1000 episodes of 100 steps | Used for model validation |
| Test | 1000 episodes of 100 steps | Used for final evaluation |
| ICL | 100 episodes of 100 steps | Used for in-context learning |
| $N$-step evaluation | 100 episodes of 100 steps each, for each $N$ value | Used for $N$-step experiments |

## D.1 DATA GENERATION PROCESS

The data generation process for both environments follows these steps:

1. Initialize the environment state randomly, ensuring a valid starting configuration.

2. For each step in the trajectory:

    (a) In the Gridworld environment, apply the dynamic update rules (e.g., moving vehicles if facing a green light).

    (b) Select a random valid action from the set of possible actions for the current state.

    (c) Apply the selected action to the environment.

    (d) Record the current state, action taken, and resulting next state.

3. Repeat step 2 for the desired number of steps (100 in our case).

For the Gridworld environment, valid actions include toggling traffic lights and performing no action. The dynamic nature of this environment means that even when no action is taken, the state may change due to vehicle movements.

For the iTHOR environment, valid actions depend on the current state and may include toggling objects (e.g., microwave, stoves), opening objects (e.g., cabinet), picking up or putting down movable objects, and performing no action.

For $N$-step experiments, we generate multiple datasets, each corresponding to a different value of $N$:

- Gridworld: We create separate datasets for $N \in \{2, 4, 6, 8\}$.

- iTHOR: We create separate datasets for $N \in \{2, 4\}$.

Each $N$-step dataset consists of 100 episodes, where each episode is created by splicing together $N$ consecutive steps from the evaluation datasets. This approach provides sequences of varying temporal lengths for our experiments.

### D.2 IN-CONTEXT LEARNING EXAMPLES

For Gridworld, we maintain a pool of 10 ICL examples, each consisting of a 3-tuple (initial_state_causal_variables, actions, end_state_causal_variables). For each iteration during training or evaluation, we randomly sample two examples from this pool to provide context for the model. This process is similar to the one employed in RAP (Hao et al., 2023).

For iTHOR, we craft fixed few-shot examples to ensure comprehensive coverage of state-action pairs. The examples are designed to demonstrate various object interactions and their outcomes. For 2-step experiments, we use 7 examples covering every state-action pair at least once. For 4-step experiments, we use 9 examples covering at least 2 of each state-action pair. This approach ensures that the model has exposure to a wide range of possible interactions within the environment.

### D.3 SELF-EVALUATION REWARDS

Following RAP (Hao et al., 2023), for the self-evaluation rewards in planning tasks, we generate samples by splicing 1-step trajectories. We select the actual action taken in the environment for "good" evaluations, providing a positive example of a correct action. For "bad" evaluations, we select a random action different from the one actually taken, providing a negative example.

## E    CRL MODEL TRAINING

This section details the training process for the Causal Representation Learning (CRL) models used in our experiments. The CRL models are trained using triplets of (state_image, text action, next_state_image) following the process described in Section 4.

### E.1 AUTOENCODER

The autoencoder is trained from scratch using 10 times more samples than the main dataset to ensure a robust representation. This approach is justified by the relative ease of obtaining unlabeled, random samples from an environment. In scenarios where this is not feasible, transfer learning from a pretrained image representation model can be employed by adding a learnable linear projection to the required dimensions and training with the original dataset size.

For the Gridworld environment, we implement an autoencoder with 40 latent dimensions and 64 hidden channels. Both the encoder and decoder consist of 2 residual blocks with SiLU activation functions. We incorporate the CoordConv operator (Liu et al., 2018) to better capture coordinate information from images. For the iTHOR environment, we employ the autoencoder architecture from BISCUIT (Lippe et al., 2023).

## E.2 NORMALIZING FLOW AND TRANSITION MODEL

For both the normalizing flow and transition model, we use the same architectures and hyperparameters as in BISCUIT (Lippe et al., 2023) as it has demonstrated strong performance in identifying causal variables from high-dimensional observations.

## E.3 TEXT ENCODER

The text encoder for the Gridworld environment is based on a pretrained Sentence Transformer (Reimers & Gurevych, 2019), specifically the all-MiniLM-L6-v2 model[3], augmented with a 2-layer MLP head with $64$ hidden dimensions. For iTHOR, we use a pretrained SigLIP model (Zhai et al., 2023)[4] with a similar 2-layer MLP head. In both cases, the pretrained encoders remain frozen during training, with only the MLP head being updated.

## E.4 TRAINING PARAMETERS

Key training parameters for each environment are as follows: For Gridworld, we use a learning rate of $3 \times 10^{-3}$ for the main model and $3 \times 10^{-3}$ for the text MLP, batch size of $384$, and train for $300$ epochs. For iTHOR, we use a learning rate of $1 \times 10^{-3}$ for the main model and $3 \times 10^{-3}$ for the text MLP, batch size of $64$, and train for $100$ epochs. Both environments employ a warmup period of $100$ steps and a sequence length of $2$ for training.

## F MODEL SELECTION

This section details our model selection procedure for the different components of our framework.

## F.1 MODEL COMPONENTS

For the text encoder, we performed 5-fold cross-validation to select the optimal hyperparameters for the MLP head architecture and training parameters. The search parameters for the planning algorithm were optimized using Bayesian optimization with 15 trials.

## F.2 SENSITIVITY

Our experiments indicated that the framework's performance is relatively robust to variations in the model training hyperparameters. The causal encoder and text encoder components showed stable performance across different configurations. However, we observed higher sensitivity to the exploration weight parameter $w$ in the search algorithm due to the interaction between exploration-exploitation trade-off and reward scaling.

## G CAUSAL MAPPER

The causal mapper $m_\theta$ extracts interpretable causal variables from the learned disentangled representations.

This process allows for a non-injective mapping from latent dimensions to causal variables. For instance, if we have a causal variable "cabinet_state", the first stage might learn that latents 1, 5, and 7 are the most predictive for this variable. In the second stage, a specific predictor would learn to map from these dimensions to either 0 (closed) or 1 (open).

The causal mapper $m_\theta$ is implemented in two stages:

## G.1 TARGET ASSIGNMENT

This stage uses a single MLP $f_{\text{assign}}$ to predict all causal variables from each latent dimension independently:

---

[3] https://huggingface.co/sentence-transformers/all-MiniLM-L6-v2
[4] https://huggingface.co/timm/ViT-B-16-SigLIP

$$\hat{C} = f_{\text{assign}}(z \odot M, M) \tag{14}$$

where $M \in \{0,1\}^L$ is a mask and $\odot$ is element-wise multiplication. $M$ is set to the identity.

For each latent dimension $i$, we create a mask $M_i$ where only the $i$-th element is 1 and the rest are 0. We then batch these masks along with the corresponding masked latent vectors:

$$\begin{bmatrix} z \odot M_1 \\ z \odot M_2 \\ \vdots \\ z \odot M_L \end{bmatrix}, \begin{bmatrix} M_1 \\ M_2 \\ \vdots \\ M_L \end{bmatrix} \tag{15}$$

This batched input is fed into $f_{\text{assign}}$, which outputs predictions for all causal variables for each masked input. The output shape is $[L, C]$, where $L$ is the number of latent dimensions and $C$ is the number of causal variables.

We then compute the correlation between these predictions and the ground truth causal variables. This allows us to identify which latent dimensions are most predictive of each causal variable. We apply a correlation threshold (in our experiments we use $0.1$) to determine which latent dimensions are relevant for each causal variable to determine each $M'_j$.

### G.2 CAUSAL PREDICTION

Individual MLPs $f_{\text{causal},j}$ are trained for each causal variable $j$, using only the relevant latent dimensions identified in stage 1:

$$\hat{C}_j = f_{\text{causal},j}(z \odot M'_j) \tag{16}$$

where $M'_j$ is the mask for causal variable $j$.

The output layer of each $f_{\text{causal},j}$ is adjusted based on the a priori known type of the causal variable (categorical, numerical, angle).

## H  STATE DESCRIPTION GENERATOR

The state description generator $s$ is responsible for converting the causal variables into a human-readable natural language description of the current state. This process can be implemented in various ways:

### H.1  STOCHASTIC AND DETERMINISTIC IMPLEMENTATIONS

The generator can operate either stochastically or deterministically, depending on the application's needs:

1. **Stochastic:** This approach uses a language model with a temperature greater than 0, which allows for a variety of possible descriptions for the same state. This variability can be useful in scenarios where diverse language outputs are desired.

2. **Deterministic:** This method involves either setting the temperature of a language model to 0, ensuring consistent outputs, or using a rule-based system that directly maps causal variables to fixed phrases or sentences.

### H.2  EXAMPLE OF STATE DESCRIPTION GENERATION

For instance, given a dictionary of causal variables as follows:

```
{
    "cabinet_state": 1,
    "light_color": 0,
    "door_angle": 45
}
```

The state description generator might produce a sentence like: "The cabinet is open. The traffic light is showing a red signal. The door is partially open at a 45-degree angle."

### H.3   CHOICE OF APPROACH

The choice between a stochastic and deterministic approach depends on the specific requirements of the task and the desired level of variability in the generated descriptions. For simplicity and consistency, in our experiments, we have opted for a rule-based deterministic state descriptor.

While rule-based generation is suitable for environments with reasonably sized state spaces, more complex environments may benefit from learned approaches. A fine-tuned sequence-to-sequence model or instruction-tuned LLM could generate natural descriptions from causal variables while maintaining consistency. The key requirement is that the mapping from causal variables to descriptions remains reliable and interpretable, allowing the planning agent to reason effectively about state transitions.

The modular nature of our framework allows for easy substitution of the state description generator. This flexibility ensures that as environments become more complex, the description generation can be adapted accordingly while maintaining the benefits of our causally-aware planning approach.

## I   ENABLING THE BASELINE TO PROCESS IMAGE STATES

To enable LLaMA to process the environment states, we implement a conversion of visual states to natural language descriptions using the ground truth causal variables. This process ensures fair comparison with our causal world model while maintaining the LLM's ability to reason about the environment.

For each initial state, we extract the ground truth causal variables and use the same rule-based state description generator employed in our causal world model to convert these variables into natural language. For example, in GridWorld, a state with causal variables `blue_car_x: 2`, `blue_car_y: 3`, `cyan_light_state: green` would be converted to "The blue car is at position (2,3). The cyan traffic light is showing green.".

The baseline LLM then uses this initial state description to reason about subsequent states and actions, relying on its world model capabilities to predict state transitions. This approach ensures that the baseline has access to the initial information as our causal world model, with the key difference being that our model learns the causal structure while the baseline relies on its pretrained knowledge for state transition predictions.

## J   CAUSAL MAPPER ANALYSIS

We present a statistical analysis framework to evaluate the performance of our causal mapper in the GridWorld environment.

### J.1   ANALYSIS METHODOLOGY

Our evaluation framework consists of two core components:

1. **Overall Performance Analysis:** We track mean absolute error (MAE) across all dimensions against training set size. Standard deviation bands are computed from three independent training runs to illustrate the variance in performance across different training instances.

2. **Dimension-wise Evolution Analysis:** We analyze how prediction accuracy for each causal variable evolves with training size using heatmaps, with darker colors indicating better performance.

Statistical significance is assessed using the criterion that standard deviation should be less than half the mean value, indicating reliable performance measurements.

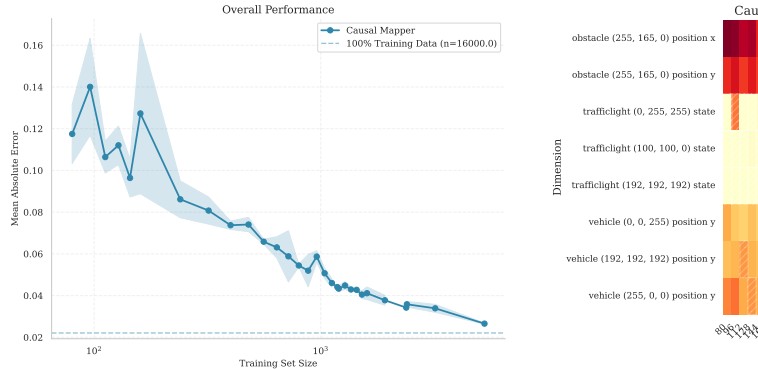
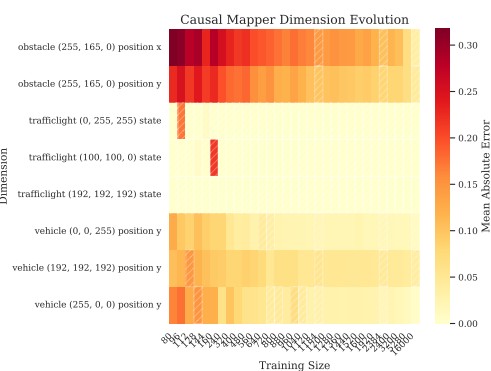

(a) Model performance (MAE) vs training size for the causal mapper in GridWorld. Error bands represent one standard deviation across three independent runs.

(b) Dimension-wise performance analysis showing the evolution of prediction accuracy across dimensions at different training sizes. Darker green colors indicate better performance.

Figure 3: Performance analysis of the causal mapper showing both overall error metrics and dimension-wise evolution.

## J.2 RESULTS

Our analysis reveals strong performance of the causal mapper approach in terms of data efficiency and prediction accuracy. The causal mapper achieves adequate accuracy (MAE < 0.05) with approximately 1200 labeled examples, demonstrating effective learning from disentangled representations.

The dimension-wise evolution analysis reveals distinct learning patterns across different types of causal variables. The causal mapper exhibits rapid early learning for traffic light states, achieving high accuracy with minimal data. For positional variables, we observe more gradual but consistent improvement as training data increases. This pattern suggests that binary state variables (like traffic light states) are easier to learn than continuous positional variables, which require understanding more complex spatial relationships.

The performance analysis shows consistent improvement across all dimensions as training size increases, with particularly strong performance in predicting traffic light states even in low-data regimes. The small standard deviation bands indicate stable learning across different training runs, suggesting robust performance regardless of initialization conditions.

## K EVALUATION METHODOLOGY

Given the stochastic nature of both the Gridworld and iTHOR environments, we have implemented specific adjustments to our evaluation methodology. These adjustments ensure that our performance metrics accurately reflect the models' understanding of the underlying causal structure while accounting for inherent randomness in the environments.

### K.1 GRIDWORLD ENVIRONMENT

In the Gridworld environment, we make the following adjustment:

- **Boulder Position Exclusion:** We exclude the boulder's position from the final state evaluation. This is because the boulder's movement is stochastic and not determined by the causal structure we aim to learn and evaluate other than the fact that it was moved or not.

**Rationale:** The boulder's position can vary due to random factors not captured in our causal model. By excluding it from our evaluation, we focus on the aspects of the environment that are causally determined by the actions and states we're modeling.

## K.2 iTHOR Environment

For the iTHOR environment, we implement a more nuanced approach:

- **Coordinate Categorization:** We categorize the $x$, $y$, and $z$ coordinates for objects with stochastic movements into discrete position categories.
- **Category-based Evaluation:** Instead of comparing exact coordinates, we check whether objects end up in the correct category of positions after an action.

**Rationale:** In iTHOR, object movements can have slight variations due to a) inherent stochasticity in movements, and b) physics simulations, even when the same action is applied. By categorizing positions, we can evaluate whether the model correctly predicts the general outcome of an action (e.g., "on the counter" vs. "in the microwave") without being overly sensitive to minor coordinate differences.

## K.3 Implementation Details

For both environments, we implement these adjustments as follows:

1. **State Representation and Prediction:** We maintain the full state representation, including all object positions and attributes, for both the actual and predicted states.
2. **Dynamic Evaluation:** During the comparison of predicted states to ground truth states, we dynamically apply our adjustment rules:
    - For Gridworld, we dynamically ignore the boulder's position when comparing states.
    - For iTHOR, we dynamically categorize the exact x, y, z coordinates into position categories (e.g., "on the counter", "in the microwave") and compare these categories instead of the exact coordinates.
3. **Accuracy Calculation:** We calculate accuracy based on the match between predicted and actual states after applying these dynamic adjustments during the comparison process.

## L COMPUTATIONAL OVERHEAD ANALYSIS

We performed detailed benchmarks comparing single-step predictions between the LLM-based world model and our CRL world model. The analysis was conducted on 5 GridWorld samples, with 10 runs per sample after warmup, using a single NVIDIA A100-40GB GPU.

Our CRL world model consists of three main components: an autoencoder (4.5M parameters), a normalizing flow (2.9M parameters), and a transition prior (28.7M parameters), totaling 36.1M parameters. For comparison, we used LLaMA 3 8B quantized to 6 bits via ExLlamaV2 as our baseline LLM world model.

The benchmarks revealed that the CRL world model achieves an average inference time of 27ms, compared to 2.2s for the LLM world model—representing an approximately 82x speedup. This computational difference has significant implications for planning tasks. For example, a 10-branch tree search would take approximately 22 seconds with LLM calls versus just 0.27 seconds with the CRL world model.

This substantial performance difference becomes particularly important in scenarios requiring multiple forward simulations or when real-time planning is necessary. The efficiency of our CRL world model enables more extensive tree searches and faster iteration during planning, while maintaining high prediction accuracy as demonstrated in our main experimental results.

# M CAUSAL WORLD MODEL ALGORITHM

This section presents the formal algorithm for sampling from/performing inference with the Causal World Model. The algorithm takes as input the trained model components and an initial state, and produces a sequence of latent states and their corresponding natural language descriptions.

---

**Algorithm 1** Inference with the Causal World Model

---

**Require:** Observation space $\mathcal{X}$, latent space $\mathcal{Z}$, action description space $\mathcal{L}$, action encoding space $\mathcal{A}$; observation encoder $e_\psi$, normalizing flow $f_\phi$, action encoder $\mathcal{L}_e$, transition model $p_\omega$, causal mapper $m_\theta$, state description generator $s$; initial observation $X_0 \in \mathcal{X}$; action descriptions $\{L^t\}_{t=0}^{T-1} \in \mathcal{L}^T$

1: **function** $\mathcal{E}(X \in \mathcal{X})$
2:      $\mathbf{E} \leftarrow f_\phi(e_\psi(X))$                 ▷ Causal encoding of observation
3:      **return** $\mathbf{E}$
4: **end function**
5: **function** ENCODEACTION$(L \in \mathcal{L})$
6:      $a \leftarrow \mathcal{L}_e(L)$             ▷ Encode action description into action latent space
7:      **return** $a$
8: **end function**
9: **function** $\mathcal{G}(z \in \mathcal{Z})$
10:     $\mathbf{C} \leftarrow m_\theta(z)$                 ▷ Map latent state to causal variables
11:     $\ell \leftarrow s(\mathbf{C})$             ▷ Generate natural language state description
12:     **return** $\ell$
13: **end function**
14: **function** SAMPLENEXTSTATE$(z_t \in \mathcal{Z}, a_t \in \mathcal{A})$
15:     $z_{t+1} \sim p_\omega(z_{t+1} \mid z_t, a_t)$             ▷ Predict next latent state
16:     $\ell_t \leftarrow \mathcal{G}(z_t)$             ▷ Generate current state description
17:     $\ell_{t+1} \leftarrow \mathcal{G}(z_{t+1})$          ▷ Generate next state description
18:     **return** $(z_{t+1}, \ell_t, \ell_{t+1})$
19: **end function**
20: **function** INFERENCETRAJECTORY$(X_0 \in \mathcal{X}, \{L^t\}_{t=0}^{T-1})$
21:     $z_0 \leftarrow \mathcal{E}(X_0)$                ▷ Initialize latent state
22:     $\ell_0 \leftarrow \mathcal{G}(z_0)$            ▷ Generate initial state description
23:     **yield** $(z_0, \ell_0)$
24:     **for** $t = 0$ to $T - 1$ **do**
25:         $a_t \leftarrow$ EncodeAction$(L^t)$          ▷ Encode action description
26:         $(z_{t+1}, \ell_t, \ell_{t+1}) \leftarrow$ SAMPLENEXTSTATE$(z_t, a_t)$
27:         **yield** $(z_{t+1}, \ell_{t+1})$
28:         $z_t \leftarrow z_{t+1}$              ▷ Update latent state for next iteration
29:     **end for**
30: **end function**

---

# N  MODIFIED MCTS PLANNING ALGORITHM

We adapt the Reasoning via Planning (RAP)(Hao et al., 2023) Monte Carlo Tree Search (MCTS) algorithm (Kocsis & Szepesvári, 2006; Coulom, 2006) for our causally-aware planning framework. Our modifications primarily focus on integrating the causal world model and leveraging its capabilities. Algorithm 2 presents our modified version of the MCTS algorithm.

---

**Algorithm 2** Causally-Aware MCTS

---

**Require:** Initial image $\mathbf{X}^0$, causal world model (Algorithm 1), LLM agent, depth limit $L$, number of roll-outs $N$, exploration weight $w$, intuition ICL samples $\mathcal{D}_{\text{ICL}}^{\text{int}}$, self-evaluation ICL samples $\mathcal{D}_{\text{ICL}}^{\text{self}}$

1: Initialize memory of actions $A : \mathcal{Z} \mapsto \mathcal{L}$, children $c : \mathcal{Z} \times \mathcal{L} \mapsto \mathcal{Z}$ and rewards $r : \mathcal{Z} \times \mathcal{L} \mapsto \mathbb{R}$
2: Initialize the state-action value function $Q : \mathcal{Z} \times \mathcal{L} \mapsto \mathbb{R}$ and visit counter $N : \mathcal{Z} \mapsto \mathbb{N}$
3: $\mathbf{z}^0 \leftarrow \mathcal{E}(\mathbf{X}^0)$, $\ell^0 \leftarrow \mathcal{G}(\mathbf{z}^0)$                          ▷ Initialize root node
4: **for** $n \leftarrow 0, \dots, N - 1$ **do**
5:      $t \leftarrow 0, \mathbf{z}^t \leftarrow \mathbf{z}^0, \ell^t \leftarrow \ell^0$
6:      **while** $N(\mathbf{z}^t) > 0$ **do**                                           ▷ Selection
7:          $N(\mathbf{z}^t) \leftarrow N(\mathbf{z}^t) + 1$
8:          $a_t \leftarrow \arg\max_{a \in A(\mathbf{z}^t)} \left[ Q(\mathbf{z}^t, a) + w \sqrt{\frac{\ln N(\mathbf{z}^t)}{N(c(\mathbf{z}^t, a))}} \right]$
9:          $r_t \leftarrow r(\mathbf{z}^t, a_t), \mathbf{z}^{t+1} \leftarrow c(\mathbf{z}^t, a_t)$
10:         $t \leftarrow t + 1, \mathbf{z}^t \leftarrow \mathbf{z}^{t+1}, \ell_t \leftarrow \mathcal{G}(\mathbf{z}^t)$
11:      **end while**
12:      **while** $\mathbf{z}^t$ is not a terminal state $\wedge\ t \leq L$ **do**
13:          $\mathcal{A}_t \leftarrow \text{GetValidActions}(\ell^t)$
14:          **for** $a \in \mathcal{A}_t$ **do**                                    ▷ Expansion
15:             $\mathbf{z}^{t+1}, \_, \ell^{t+1} \leftarrow \text{SAMPLENEXTSTATE}(\mathbf{z}^t, \mathcal{L}_e(a))$     ▷ Use the Causal World Model
16:             $r_{\text{intuition}} \leftarrow -\log p_{\text{LLM}}(a \mid \ell^t, \mathcal{D}_{\text{ICL}}^{\text{int}})$
17:             $r_{\text{self-eval}} \leftarrow -\log p_{\text{LLM}}(\text{“good”} \mid \ell^t, a, \mathcal{D}_{\text{ICL}}^{\text{self}})$
18:             $r(\mathbf{z}^t, a) \leftarrow r_{\text{intuition}} + r_{\text{self-eval}}$
19:             Update $A(\mathbf{z}^t) \leftarrow A(\mathbf{z}^t) \cup \{a\}, c(\mathbf{z}^t, a) \leftarrow \mathbf{z}^{t+1}$
20:          **end for**
21:          $a_{t+1} \leftarrow \arg\max_{a \in A(\mathbf{z}^t)} r(\mathbf{z}^t, a)$                             ▷ Simulation
22:          $r_t \leftarrow r(\mathbf{z}^t, a_{t+1}), \mathbf{z}^{t+1} \leftarrow c(\mathbf{z}^t, a_{t+1})$
23:          $t \leftarrow t + 1, \mathbf{z}^t \leftarrow \mathbf{z}^{t+1}, \ell^t \leftarrow \mathcal{G}(\mathbf{z}^t)$
24:      **end while**
25:      **for** $t' \leftarrow t, \dots, 0$ **do**                                     ▷ Back propagation
26:          Update $Q(\mathbf{z}^{t'}, a_{t'})$ with $\{r_{t'}, r_{t'+1}, \dots, r_t\}$
27:      **end for**
28: **end for**

---

The key modifications in our algorithm compared to the original RAP MCTS are:

1. **State Representation**: We use disentangled causal latent representations $\mathbf{z}$ for states, starting from an encoded initial image $\mathbf{X}^0$ (line 3).

2. **Causal World Model Integration**: We employ our trained causal world model (Algorithm 1) to predict the next state and generate state descriptions (line 15).

These modifications allow our MCTS algorithm to leverage the causal understanding provided by the causal world model, while also incorporating the strengths of the LLM agent for action selection and evaluation. The use of disentangled latent representations $\mathbf{z}$ allows for efficient and robust state transitions, while the natural language descriptions $\ell$ enable interaction with the LLM agent.

While our current implementation uses a predefined set of valid actions, the framework could potentially be extended to sample actions directly from the LLM for open-ended domains where the action space is not easily enumerable.

