# OpenReview forum: "Language Agents Meet Causality -- Bridging LLMs and Causal World Models"
_ICLR.cc/2025/Conference — ICLR 2025 Poster_

### Official Review · Reviewer_ZjUs · 2024-10-22

**Soundness:** 3
**Presentation:** 3
**Contribution:** 3
**Rating:** 8
**Confidence:** 4

**Summary:**

The author proposed to introduce causal representation learning (CRL) to better understand the causal structure of a environment, and they incorporate such CRL with LLM to enable causally-aware planning by mapping causal representations to natural language.

**Strengths:**

The paper proposes to introduce CRL for world model to better understand the causal structure of environments. This intuitive idea provides a solid future approach to further improve the performance of world models, which is essential to build with intelligent agent with System 2 reasoning abilities.
The author analyzes the underlying math principles of the proposed CRL model and shows with experiments of the performance in metrics of comparison between induced state variables and ground-truth, and multi-step high-level accuracy of the predicted states. The experiments show large performance increase than baseline pure language model.
The author also use the result causal world model to do experiments on synthetic planning tasks. The results show consistent superior performance to baseline LM, especially on longer planning horizons.

**Weaknesses:**

+ It's not clear how much the text decoded from CRL model helps the LLM reasoning. It would be clearer if the author can show some examples and/or conduct ablation experiment on this. (also see questions)
+ The experiments compare **CB**, **TB**, and **HB** are based on the low-data scenarios, but the decision to use **TB** in subsequent experiments is for 100% data. This decision lacks a continuity of experiment settings, and it's unclear which setting would be the best for subsequent experiments. (also see questions)
+ The paper is a good starting of causal world model studies, but I have to mention that the experiment environment is synthetic and relatively easy. It would be more fascinating to use more general environments, and there leaves doubt whether the method can be easily scaling up to more general or even real world scenarios.

**Questions:**

+ Do we have examples of decoded text (from CRL to LLM)? Is that human-readable? Is there any possible insights with analysis of the decoded text?
+ (Clarification) In my understanding, the performance increase comes from (1) the text decoded from CRL makes it easier for LLM to reason and plan (2) the CRL model has a higher accuracy in predicting future states. If my understanding is correct, do we have ablation experiments on the effect of the two aspects?
+ (Clarification) Does the Baseline LM refer to RAP + LLaMA 3 or LLaMA3 with other prompting methods?
+ In Table 1, **TB** outperforms **HB** starting from 1.0%, outperforming **HB** much at **1.5%**, but performs weaker than **HB** in 100%. What are the trend between 1.5% and 100% subsample rate? Does it mean the necessity of **TB** decreases when the data scales up enough?
+ The **HB** settings shall contain all information in the **TB** settings. Why would the model performs worse than **TB** with increased subsample percentage? Since **HB** performs better in 100% ($10^6$ image states), would it be better to use **HB** in subsequent experiments?
+ Can you include SNA (success weighted by number of actions) in the planning results? It's a common metric to combine success and # of steps.
+ (Typo) Page 4 footnote, prioi -> prior
+ (Typo) Figure 1, missing an arrow from $\bf C^1$ to Text Decoder.

---

> ### Author Response · Authors · 2024-11-20
>
> Thank you to Reviewer ZjUs for the comprehensive feedback. Your detailed questions about text representation and experimental design have helped us clarify the presentation.
>
> ---
>
> ### **Weakness:**
> > "It's not clear how much the text decoded from CRL model helps the LLM reasoning."
>
> The decoded text is human-readable and follows a consistent format. Examples include "The blue car is at position (2,3)" and "The cyan traffic light is green." The text format enables the LLM to leverage its pre-trained knowledge while maintaining the interpretability of the state representations. This allows for easier debugging and verification of the model's reasoning process (also see the answer to the corresponding question).
>
> ---
>
> ### **Weakness:**
> > "The experiments compare CB, TB, and HB are based on the low-data scenarios, but the decision to use TB in subsequent experiments is for 100% data. This decision lacks a continuity of experiment settings, and it's unclear which setting would be the best for subsequent experiments."
> ### **Question:**
> >"Why would [HB] perform worse than TB with increased subsample percentage? Since HB performs better in 100% (10^6 image states), would it be better to use HB in subsequent experiments?"
>
> Section 6.1 is intended to serve two purposes: (1) demonstrating that TB representations are effective for CRL, and (2) showing they outperform coordinate-based representations in low-data regimes. We first establish that text-based representations are effective and then use the strongest CRL module from those. Thus, for planning, text-based representations are preferred as coordinate outputs would be dissimilar to the LLM's pretraining distribution. Given enough data, the evaluated representations converge to practically indistinguishable performance.
>
> Additionally, our text-based representations offer important advantages:
>
> - **Robustness to paraphrasing** – different descriptions of the same action map to the same latent space region.
> - **No need to enumerate all possible actions across environments.**
> - **Independence from specific action encoding formats** used during pre-training (e.g., third-person vs first-person descriptions, varying adjective usage).
>
> These properties arise from our language encoder backbone's ability to map varied textual descriptions to consistent latent representations, making our approach more practical for real-world deployment. While HB performs well with full data for pure representation learning, TB is more suitable for the complete framework due to its natural alignment with LLM capabilities.
>
> ---
>
> ### **Weakness:**
> > "The paper is a good starting of causal world model studies [...] but the experiment environment is synthetic and relatively easy. It would be more fascinating to use more general environments [...] or even real world scenarios."
>
> Our framework is designed to benefit from advances in CRL, which is a rapidly developing field. As stronger CRL methods are being developed for handling more complex environments, they can be directly integrated through our modular design.
>
> ---
>
> ### **Question:**
> > "Do we have examples of decoded text?"
>
> Yes, the decoded text is human-readable and interpretable. For example, it might output: "The blue car is at position (2,3), while the traffic light is green, …". This provides insights for failure analysis by allowing us to discern whether errors stem from incorrect state representations.
>
> ---
>
> ### **Question:**
> > "(Clarification) In my understanding, the performance increase comes from (1) the text decoded from CRL makes it easier for LLM to reason and plan (2) the CRL model has a higher accuracy in predicting future states. If my understanding is correct, do we have ablation experiments on the effect of the two aspects?"
>
> The performance gain stems from (2). Both approaches use identical state representations, achieved by maintaining consistent text formatting in the in-context learning examples for our method and the baseline. The LLM in the baselines then adheres to the same format when generating state representations. This makes it possible to attribute the performance improvements to the CRL world model's superior state prediction accuracy. We clarified this in lines 521–526.
>
> ---
>
> ### **Question:**
> > "(Clarification) Does the Baseline LM refer to RAP + LLaMA 3 or LLaMA 3 with other prompting methods?"
>
> It refers to RAP+LLaMA 3, now clarified on line 353 and in tables 2, 3, and 4. Direct prompting for plan generation performed significantly worse, even with additional context examples. Our framework is orthogonal to the specific search algorithm used.
>
> ---
>
> cont.

---

> > ### Author Response · Authors · 2024-11-20
> >
> > ### **Question:**
> > > "Can you include SNA (success weighted by number of actions) in the planning results?"
> >
> > While we cannot compute SNA due to lacking shortest path information in our data generation process, our results demonstrate that both models frequently achieve goals in fewer steps than the N-step upper bound. This "sub-N performance" occurs through two mechanisms we discuss in Section 6.3: action cancellation in static environments (e.g., toggling an object twice returns to the initial state) and both action cancellation and dynamic state changes in GridWorld (e.g., vehicle movements, toggling irrelevant traffic lights twice). The causal world model consistently finds efficient paths, indicating a better success-to-steps ratio even without explicit SNA calculations.

---

> ### Comment · Reviewer_ZjUs · 2024-11-21
> **Feedback to Author Rebuttal**
>
> Thanks for clarifying most problems I had. I suggest the author to include some text decoded from CRL model to help understand the framework and show the human-readable nature of them, maybe in figures or appendix.
>
> I understand the point from author about using TB representations, and understand the limitation from data generation preventing SNA calculation.
>
> Most of my concerns have been addressed. I would like to keep my score.

---

### Official Review · Reviewer_jsxQ · 2024-11-01

**Soundness:** 3
**Presentation:** 4
**Contribution:** 4
**Rating:** 6
**Confidence:** 2

**Summary:**

The paper proposes a novel framework that integrates Causal Representation Learning (CRL) with Large Language Models (LLMs) to enhance reasoning and planning capabilities in interactive environments. The framework utilizes a causal world model that maps high-dimensional state representations to causal variables, which are linked to natural language expressions. This integration allows the LLM to interact with the causal world model, effectively simulating multiple possible futures before taking actions. The approach is evaluated on causal inference and planning tasks across different environments and temporal scales, demonstrating that the causally-aware method outperforms traditional LLM-based reasoners, especially for longer planning horizons.

**Strengths:**

The exploration of text-based action representations is particularly interesting, as it shows potential advantages in low-data regimes. I particularly liked the connection to the RL decision making problem. The results indicate that the proposed framework can potentially improve performance in causal inference and planning tasks, which is valuable for the broader ICLR community.

**Weaknesses:**

Please see the questions

**Questions:**

Can the authors elaborate on how the framework handles situations where the causal factors are not easily identifiable or where the causal relationships are highly complex?

Following on the above point, how were the parameters chosen for the various components of the framework, and how sensitive are the results to these parameters?

---

> ### Author Response · Authors · 2024-11-20
>
> We value Reviewer jsxQ's questions about framework limitations, which helped us better articulate our parameter selection process and robustness considerations.
>
> ---
>
> ### **Question:**
> > "Can the authors elaborate on how the framework handles situations where the causal factors are not easily identifiable or where the causal relationships are highly complex?"
>
> When the CRL component correctly identifies the causal factors, our framework enables effective reasoning and planning, as demonstrated by our results. In cases where causal identification is challenging, the framework's performance would be limited by the upstream CRL component's capabilities.
>
> ---
>
> ### **Question:**
> > "How were the parameters chosen?"
>
> The hyperparameters were selected as follows, now detailed in Appendix F:
>
> - **Causal encoder:** We use the hyperparameters from the BISCUIT paper.
> - **Text encoder:** Cross-validation across 5 folds.
> - **Search parameters:** Bayesian optimization with 15 trials.
>
> Our experiments throughout the project indicated low sensitivity to model training hyperparameters but higher sensitivity to the search exploration parameter due to reward scaling.

---

> > ### Author Response · Authors · 2024-11-23
> >
> > Dear Reviewer jsxQ,
> >
> > As we near the end of this discussion period, we would like to thank you once again for your valuable feedback. In our rebuttal, we addressed the weaknesses you and the other reviewers highlighted – such as clarifying how hyperparameters have been selected, and emphasizing that the CRL approach is actually faster than RAP – and made revisions to the paper accordingly. Following this, we have made substantial revisions to the paper, which we believe have significantly improved its quality.
> >
> > We would greatly appreciate it if you could let us know whether our responses adequately address your concerns.

---

> > > ### Comment · Reviewer_jsxQ · 2024-11-24
> > >
> > > Thank you for adding Appndx F. I raised my score for presentation and contribution.

---

### Official Review · Reviewer_wNeN · 2024-11-03

**Soundness:** 3
**Presentation:** 3
**Contribution:** 3
**Rating:** 6
**Confidence:** 3

**Summary:**

The paper proposed to learn a causal world model from a sequence of states and actions, where the states are images and actions are described in natural language. The proposed method builds upon BISCUIT and proposes representing the actions in natural language. The paper conducted experiments in GridWorld and iTHOR environment and the R^2 scores of the learned representation are claimed to be higher in low-data regimes. Experiments also show that the learned causal world model is more accurate at predicting the next state, and improves planning performance when compared to a general LLM (Llama 3).

**Strengths:**

- The paper proposed a method to learn a causal world model from a sequence of image states and text action descriptions, and demonstrated superior performance in the accuracy of the learned world model.
- Presentation of the paper is clear and easy to follow.

**Weaknesses:**

1. Some components of the framework would not be available in more realistic environments, e.g. 1) a set of annotated images with ground-truth causal variables is used in training, which is likely not available as we may not know the causal variables for more realistic environments; 2) a rule-based state description generator may not be available for complex environments where we don't know what are the true causal variables.
2. Given the simplicity of the environments, and that the proposed method is trained on these particular domains while the baseline is a general LM, the superior performance of the learned causal world model is less convincing. I would suggest comparing with a supervised fine-tuned version of the baseline LM. Since the proposed method uses a set of annotated images to train the causal mapper, supervised fine-tuning is possible with these annotated data: use the state description generator to convert the ground-truth causal variables of the states to natural language, and we can obtain a sequence of natural language action and natural language description of the states.

**Questions:**

1. How is coordinate-based action representation implemented? A 2-d vector, or simple text like “(2,3)” encoded by the encoder?
2. What does HB action representation look like and how does it differ from TB? An example of CB, TB, and HB would help to explain.
3. line 377, “better sample efficiency” would allow TB to perform well in extremely low-data scenarios as well, the reviewer think the experimental results doesn’t support the claim of “better sample efficiency” of TB.
4. In table 2, the baseline language model and the causal world model both predicts the next state in natural language, how is it evaluated against the ground-truth next state?
5. How big is the set of annotated images used to train the causal mapper m_\theta?

---

> ### Author Response · Authors · 2024-11-20
>
> We thank Reviewer wNeN for their thorough analysis. Your inquiries about action representations and experimental design have helped us bolster our presentation.
>
> ---
>
> ### **Weakness:**
> > "Some components of the framework would not be available in more realistic environments, e.g. 1) a set of annotated images with ground-truth causal variables is used in training, which is likely not available as we may not know the causal variables for more realistic environments; 2) a rule-based state description generator may not be available for complex environments where we don't know what are the true causal variables."
>
> Our framework requires annotation only of causal variables and their values (e.g., object positions, light states), not their relationships or temporal dependencies. This is significantly easier than annotating temporally consistent sequences of states with their corresponding actions. The causal mapper requires a small labeled set (6.25% of validation data: 1000 frames for GridWorld, 1500 for iTHOR), used only for the decoder phase. We added an analysis of the mapper’s data efficiency to the paper (see Appendix J).
>
> While we used a rule-based state description generator for simplicity, this component is modular and can be replaced with a sequence-to-sequence model or fine-tuned LLM for more complex environments. We have extended Appendix H.3 to detail these alternatives. The key requirement is consistency in the mapping from causal variables to descriptions, achievable through various approaches.
>
> ---
>
> ### **Weakness:**
> > "Given the simplicity of the environments, and that the proposed method is trained on these particular domains while the baseline is a general LM, the superior performance of the learned causal world model is less convincing. I would suggest comparing with a supervised fine-tuned version of the baseline LM. Since the proposed method uses a set of annotated images to train the causal mapper, supervised fine-tuning is possible with these annotated data."
>
> Thank you for the suggestion! Fine-tuning the baseline LM would require triples of state, action, and next state in *natural language* to learn the causal dynamics. This is a different experimental setup that makes a stronger assumption on the available annotations than our method. The labeled examples we use for the causal mapper are unordered individual frames (clarified on lines 284-286), lacking the temporal structure needed to learn causal dynamics. Our approach learns the causal dynamics from the environment with CRL.
>
> In addition, the benefits of supervised fine-tuning could be limited because:
>
> 1. **2-9 planning problem solutions spanning multiple steps are already used for in-context learning depending on the experiment**, detailed in Appendix D.2.
> 2. **The pre-prompt and few-shot examples already provide environment-specific information.**
>
> ---
>
> ### **Question:**
> > "How is coordinate-based action representation implemented? A 2-d vector, or simple text like '(2,3)' encoded by the encoder?"
>
> It uses fixed 2D sinusoidal encodings mapping 2D inputs to a higher-dimensional space using high-frequency functions, similar to the non-learnable components in NeRF/Fourier feature encodings [1, 2]. This is now clarified in lines 328-333.
>
> ---
>
> ### **Question:**
> > "What does HB action representation look like and how does it differ from TB? An example of CB, TB, and HB would help to explain."
>
> We have added examples on lines 328-333:
>
> - **CB:** `(2,3)` → sinusoidal encoding
> - **TB:** `"move two steps right and three steps up"` → text embedding
> - **HB:** concatenation of both above representations
>
> ---
>
> ### **Question:**
> > "line 377, 'better sample efficiency'..."
>
> Thank you for this observation. We have weakened the claim about sample efficiency on line 377.
>
> ---
>
> ### **Question:**
> > "In table 2, how is it evaluated against the ground-truth next state?"
>
> We compare the predicted states against ground truth by focusing on causally relevant aspects while accounting for environmental stochasticity. For instance, in iTHOR, we categorize object positions into discrete regions (e.g., “on counter,” “in microwave”) rather than comparing exact coordinates, since physics-based interactions can lead to slight positional variations even for identical actions. This ensures we evaluate meaningful causal predictions rather than exact numerical matches. Full evaluation details are provided in Appendix K.
>
> ---
>
> ### References
>
> [1] Tancik, M. et al. (2020) *‘Fourier features let networks learn high frequency functions in low dimensional domains’,* Advances in Neural Information Processing Systems, 33, pp. 7537–7547.
>
> [2] Mildenhall, B. et al. (2020) *‘NeRF: Representing Scenes as Neural Radiance Fields for View Synthesis’,* in ECCV.

---

> > ### Author Response · Authors · 2024-11-23
> >
> > Dear Reviewer wNeN,
> >
> > As we approach the end of the discussion period, we would like to thank you once again for your valuable feedback. In our rebuttal, we aimed to address the weaknesses you and the other reviewers highlighted. Following this, we have made substantial revisions to the paper, which we believe have significantly improved its quality.
> >
> > We would appreciate it if you could let us know whether our responses properly address your concerns.

---

> > > ### Comment · Reviewer_wNeN · 2024-11-24
> > >
> > > Thank you for providing clarifications on action representations and evaluation details. While I'm still worried about the applicability of the proposed method on more complex scenarios with no simple known causal variables such as object positions and light states, I agree that the proposed method is not asking too much on annotation as it does not require temporally consistent sequences of states, which is a key strength of the proposed method. In light of this, I'm increasing my score to 6.

---

### Official Review · Reviewer_Uz31 · 2024-11-04

**Soundness:** 2
**Presentation:** 2
**Contribution:** 3
**Rating:** 6
**Confidence:** 4

**Summary:**

The paper introduces a framework that combines LLMs with CRL to improve reasoning and planning tasks. The framework leverages both LLMs' common sense knowledge and CRL's "look-ahead" ability on causal structures in environments. During inference, LLMs use a causal world model as a simulator for generating and processing action and state descriptions in natural language. This can be well integrated with MCTS algorithm from previous work. Tests on causal inference and planning tasks reveal that this causally-aware approach outperforms traditional LLM methods in complex, long-term planning scenarios.

**Strengths:**

- The integration of CRL and LLM planning is novel and interesting, it is straight forward to integrate it with multiple other LLM-based search algorithms, not only RAP.
- The paper investigates the form of action representation in the casual world model, and provides detailed results on them.

**Weaknesses:**

-  Reasoning via Planning is not the sota method, [1] combines LLM as a world model and LLM as a policy model, with MCTS, should be better than RAP
- Though not discussed in RAP, other literature [2] [3] suggest increasing the number of LLM calls will substantially improve the search algorithm results. How is the overhead and accuracy of LLM+CRL planning vs. increasing LLM calls with tree-search or iterative prompting?
- I'm concerned about the ability of the system to work on more complex visual environment. For example, if the accuracy of the casual world model on look-ahead is limited, we should still rely on executing the actions in the real simulator and extract observation there. Related to this concern, a comparison with RAP+the real simulator execution (some oracle) is missing.

[1] Zhao et al., Large Language Models as Commonsense Knowledge for Large-Scale Task Planning, 2023

[2] Yao et al., Tree of Thoughts: Deliberate Problem Solving with Large Language Models, 2023

[3] Zhang et al., ReST-MCTS*: LLM Self-Training via Process Reward Guided Tree Search, 2024

**Questions:**

Questions:
- How do you make sure the objects in the image X are identified with the Auto Encoder? if some objects are missing the casual model cannot run properly right?
- How does the baseline RAP work with LLama? LLama 8B does not take vision input.
- L285: the description is confusing, why causal mapper is trained using image input?
- Is LLM baseline just RAP? should just say RAP it in the table. Baseline LM is super confusing, there are many other simpler LLM prompting baselines in planning.

Typos:
- L269: p_\phi(E_t | z_t) not p_\phi(X_t | z_t)

---

> ### Author Response · Authors · 2024-11-20
>
> We appreciate Reviewer Uz31's detailed feedback, which helped us clarify our framework's relationship to existing planning methods and computational advantages.
>
> ### **Weakness:**
> > "Reasoning via Planning is not the sota method, [1] combines LLM as a world model and LLM as a policy model, with MCTS, should be better than RAP"
>
> We chose RAP as a representative baseline to isolate the impact of causal understanding, as our framework is agnostic to the specific planning algorithm used. We have clarified the statements on RAP’s classification as SoTA and our framework’s flexibility with respect to the search algorithm (lines 353-356).
>
> ---
>
> ### **Weakness:**
> > "[...] [2] [3] suggest increasing the number of LLM calls will substantially improve [RAP’s] results. How is the overhead and accuracy of LLM+CRL planning vs. increasing LLM calls with tree-search or iterative prompting?"
>
> We performed benchmarks comparing single-step predictions between the LLM-based world model (LLaMA 3 8B via ExLlamaV2 quantized to 6 bits) and the CRL world model on 5 GridWorld samples, with 10 runs per sample after warmup on a single NVIDIA A100-40GB GPU. The CRL module – totaling 36.1M parameters – achieved 27ms/inference compared to the LLM's 2.2s/inference; an ~82x speedup. This computational difference has significant implications for planning: a 10-branch tree search takes ~22s with LLM calls versus ~0.27s with CRL. Results are detailed in Appendix L.
>
> ---
>
> ### **Weakness:**
> > "I'm concerned about the ability of the system to work on more complex visual environments [...] we should still rely on executing the actions in the real simulator and extract observation there"
>
> Assuming access to an accurate simulator is a strong requirement, feasible only in controlled, simulated environments. In real-world settings, this would equate to executing actions directly in the environment, where some actions are irreversible (e.g., once an egg is cracked, it cannot be reverted to make a hard-boiled egg). Our method learns the causal dynamics to avoid performing irreversible actions in the world. Furthermore, CRL is a rapidly advancing field, and our framework will directly benefit from these advances through its modular design. As stronger CRL methods emerge for handling complex environments, they can be integrated via the causal encoder component.
>
> ---
>
> ### **Question:**
> > "How do you make sure the objects in the image X are identified with the Auto Encoder?"
>
> Our method shares the same assumptions as BISCUIT regarding the autoencoder's ability to capture objects in its representation. Specifically, we assume the autoencoder can achieve perfect reconstruction, which is a common assumption in CRL methods [4, 5, 6, 7, inter alia]. If the autoencoder fails to capture some objects, this negatively affects downstream performance. Nevertheless, the autoencoder module is orthogonal to our method and can be replaced with stronger alternatives if needed, now clarified on lines 272-275. In practice, while perfect reconstruction might not be achievable, modern architectures can achieve high-quality reconstructions sufficient for learning meaningful representations, as demonstrated by our empirical results.
>
> ---
>
> ### **Question:**
> > "How does the baseline RAP work with LLama? LLama 8B does not take vision input."
>
> To enable LLaMA to process the environment, we provide it with a language description of the ground truth causals for the initial state (using the rule-based state description generator with the ground truth causal variables of the state as input). This experimental setup is now detailed in Appendix I.
>
> ---
>
> ### **Question:**
> > "L285: why causal mapper is trained using image input?"
>
> The causal mapper is trained on the output of our causal encoding pipeline (autoencoder and normalizing flow). That is, the training data of the causal mapper are pairs ($\mathbf{z}$, $\mathbf{C}$), where $\mathbf{z}$ is a vector of the identified and separated causal variables from the normalizing flow and $\mathbf{C}$ are the ground truth causal variables. We have clarified this process on lines 284-286.
>
> ---
>
> ### **Question:**
> > "Is LLM baseline just RAP? Should just say RAP it in the table."
>
> Yes, the baseline is RAP, thanks for the suggestion. Now clearly stated in tables 2, 3, and 4.
>
> ---

---

> > ### Author Response · Authors · 2024-11-20
> >
> > ### References
> >
> > [4] Bohdan Kivva, Goutham Rajendran, Pradeep Ravikumar, and Bryon Aragam. *Identifiability of Deep Generative Models Without Auxiliary Information.* In Advances in Neural Information Processing Systems (NeurIPS), volume 35. Curran Associates, Inc., 2022.
> >
> > [5] Sebastien Lachapelle, Divyat Mahajan, Ioannis Mitliagkas, and Simon Lacoste-Julien. *Additive Decoders for Latent Variables Identification and Cartesian-Product Extrapolation.* In Advances in Neural Information Processing Systems (NeurIPS), volume 36. Curran Associates, Inc., 2023.
> >
> > [6] Sébastien Lachapelle, Pau Rodríguez López, Yash Sharma, Katie Everett, Rémi Le Priol, Alexandre Lacoste, and Simon Lacoste-Julien. *Nonparametric Partial Disentanglement via Mechanism Sparsity: Sparse Actions, Interventions and Sparse Temporal Dependencies.* arXiv preprint arXiv:2401.04890, 2024.
> >
> > [7] Johann Brehmer, Pim de Haan, Phillip Lippe, and Taco Cohen. *Weakly Supervised Causal Representation Learning.* In Advances in Neural Information Processing Systems (NeurIPS), volume 35. Curran Associates, Inc., 2022.

---

> > > ### Author Response · Authors · 2024-11-23
> > >
> > > Dear Reviewer Uz31,
> > >
> > > As we near the end of this discussion period, we would like to thank you once again for your valuable feedback. In our rebuttal, we addressed the weaknesses you and the other reviewers highlighted – such as clarifying that the CRL approach is actually faster than RAP – and made revisions to the paper accordingly.
> > >
> > > We believe your insights have significantly improved the quality of our work.
> > >
> > > We would greatly appreciate it if you could let us know whether our responses adequately address your concerns.

---

> > > > ### Comment · Reviewer_Uz31 · 2024-11-24
> > > >
> > > > Thanks to the authors for the reply. The author addressed most of my concerns. Though I still think a perfect visual reconstruction is a too strong assumption (esp for complex env), as it is a common CRL issue, I don't think it's a hard negative of the paper.
> > > >
> > > > Overall, I believe the merits of the paper outweigh its flaws. I'll keep my score.

---

### Author Response · Authors · 2024-11-20
**Common Responses**

We thank the reviewers for their constructive feedback\! We revised our paper considering the comments. Changes are highlighted in blue.

## **CR1. Strengths**

We thank the reviewers for their insightful comments and constructive feedback. Reviewer Uz31 highlighted the novelty of integrating CRL with LLM planning and its straightforward extension to multiple LLM-based search algorithms. Reviewer wNeN emphasized the superior performance of our causal world model and the clarity of presentation. Reviewer jsxQ noted the interesting exploration of text-based action representations and their advantages in low-data regimes, particularly appreciating the connection to RL decision-making. Reviewer ZjUs praised the paper's contribution to System 2 reasoning capabilities and the strong empirical results, especially in longer planning horizons.

## **CR2. Scalability to More Complex Environments**

Several reviewers (Uz31, wNeN, ZjUs) expressed concerns about the simplicity of the environments used in our experiments and the applicability of our framework to more complex or realistic settings.

Our approach is designed to be modular, allowing it to integrate advancements in causal representation learning (CRL) and thereby adapt to more complex environments. The framework's key components \- autoencoder, CRL module, and search algorithm \- can each be independently upgraded as stronger alternatives become available. The autoencoder can be replaced with more sophisticated visual encoders, the CRL module can incorporate emerging methods for causal representation learning in realistic environments, and the planning component can adopt advanced search strategies. This modularity ensures our framework can scale with advances in each component field while maintaining its core benefits in planning and reasoning.

---

### Meta-Review · Area_Chair_zUnu · 2024-12-20

**Metareview:**

This paper presents a method that combines LLMs and causal representation learning to improve reasoning and planning. The method essentially learns a world model, in which causal variables can be expressed with natural language, providing an interface with LLMs.   Experiments on causal inference and planning tasks showed strong empirical results, validating the method.

Reviewers agree that the paper is technically solid and well-written. The integration of causal world models and LLMs with the language interface is novel and works well based on the experiments. The main concern is whether the method can work in more complex environments. However, this is the first paper that proposes such novel integration and the results in the current experiments are quite convincing. Thus, the current submission is still a good contribution to the field.

**Additional Comments On Reviewer Discussion:**

Two reviewers who were originally recommending reject raised scores to 6 after the rebuttal. Reviewers reached the consensus of recommending accept.

---

### Decision · Program_Chairs · 2025-01-22

Accept (Poster)